# Template Learning: Deep learning with domain randomization for particle picking in cryo-electron tomography

Mohamad Harastani[1,2] ✉, Gurudatt Patra[1], Charles Kervrann [3] & Mikhail Eltsov [1] ✉

Cryo-electron tomography (cryo-ET) enables three-dimensional visualization of biomolecules and cellular components in their near-native state. A key challenge in cryo-ET data analysis is particle picking, often performed by template matching, which relies on cross-correlating tomograms with known structural templates. Current deep learning-based methods improve accuracy but require labor-intensive annotated datasets for supervised training. Here, we present Template Learning, a technique that combines deep learning accuracy with the convenience of training on biomolecular templates via domain randomization. Template Learning automates synthetic dataset generation, modeling molecular crowding, structural variability, and data acquisition variation, thereby reducing or eliminating the need for annotated experimental data. We show that models trained using Template Learning, and optionally fine-tuned with experimental data, outperform those trained solely on annotations. Furthermore, Template Learning provides higher precision and more uniform orientation detection than template matching, particularly for small non-spherical particles. Template Learning software is open-source, Python-based, and GPU/CPU parallelized.

Recent technological advances in cryo-electron tomography (cryo-ET) expanded our capabilities to explore the three-dimensional (3D) cell architecture at the molecular scale. On one side, the automation of vitreous cryo-lamellae milling significantly enhanced the throughput of tomographic sample preparation[1–3]. On the other side, improvements in the hardware of transmission electron microscopes (TEM), especially new generations of direct electron detectors, enabled the routine acquisition of tomograms with the resolution that allows identification of biomolecules and determination of their structure and its variations, directly in their functional environment[4–8].

The cryo-ET pipeline begins with the vitrification of the sample by plunge or high-pressure freezing. In the case of bulky samples such as eukaryotic cells and tissues, this step is followed by thinning using cryo-focused ion beam milling or cryo-sectioning. A series of 2D images (micrographs) is then obtained by rotating the sample in cryo-TEM. The resulting images, referred to as a tilt series, are computationally reconstructed into a 3D volume called a tomogram[9,10].

Although cryo-ET allows the acquisition of high-resolution data from cellular samples, it faces serious limitations. Existing sample preparation and cryo-TEM setups limit the tilting range, typically within ±60 degrees. This angular limitation leads to a missing wedge region in Fourier space, resulting in severe real-space data anisotropies commonly referred to as missing wedge artifacts[11–14]. In addition, the sensitivity of biological samples to radiation damage requires image acquisition with a limited electron dose, resulting in a compromised signal-to-noise ratio (SNR).

[1]Department of Integrated Structural Biology, Institute of Genetics and Molecular and Cellular Biology, Illkirch, France. [2]Institut Pasteur, Université Paris Cité, Image Analysis Hub (IAH), Paris, France. [3]Inria Center at University of Rennes, SAIRPICO Team, Cellular and Chemical Biology Unit, U1143 INSERM, UMR3666 CNRS, Institut Curie, PSL Research University, Campus universitaire de Beaulieu, Rennes Cedex, France. ✉e-mail: mohamad.harastani@pasteur.fr; mikhail.eltsov@igbmc.fr

Locating copies of biomolecules in cryo-electron tomograms, known as particle picking, is important for understanding biomolecular distribution in cells and constitutes an initial stage in obtaining higher resolution 3D structures via averaging and classification[15,16]. However, manual particle picking is labor-intensive, further complicated by the challenges posed by the missing wedge and low SNR.

A well-established method to automate particle picking using the similarity with an external template via cross-correlation is called template matching[17,18]. In recent years, several deep learning-based methods have emerged and were shown capable of outperforming template matching, for some studies, in speed and accuracy[19–24].

A specialized track of the SHape REtrieval Contest (SHREC) was launched to compare various deep learning methods for cryo-ET particle picking[25]. These evaluations were conducted on simulated datasets, where tomograms with ground truth annotations are available and are used for training. However, on experimental datasets, supervised deep learning methods will require a training dataset, usually obtained via manual template matching-assisted picking.

Open databases for 3D structural data of biomolecules, e.g., the Protein Data Bank (PDB)[26], give access to structures that can serve as templates for template matching. However, despite the remarkable advancements in the physical modeling of cryo-ET data[27–30], templates were not used for supervised training of deep learning models on customized particle picking[20,25,31]. Some methods found a use for simulated datasets in training deep learning-based general-purpose particle picking (e.g., cryoYOLO[32]) or unsupervised structural mining (e.g., TomoTwin[31]). However, these methods face several limitations, including molecular crowding, structural variations, or computational complexity[22,23,31,33].

Using simulations derived from previously defined or inferred structures, such as those obtained from in vitro or in silico studies, to train models to annotate similar structures in cryo-electron tomograms would be highly advantageous over the time-consuming initial annotation directly from in situ data. This raises the question of why such methodologies are not more widespread. The literature points to a challenge known as the "synthetic-to-real domain gap"[34] where deep learning models trained on simulations can only operate in the synthetic domain. This domain is characterized by attributes like texture, illumination, noise, background, and other factors, which are notably different from those in the real-world domain[34].

Domain randomization is a state-of-the-art approach for addressing the domain gap during synthetic dataset generation, specifically in the context of training models for object classification and segmentation. Domain randomization is based on simulating non-realistic scenarios that were proven powerful to supervised deep learning of models capable of generalizing to real-world data[35,36]. It hinges on three principles: (1) utilizing models of target objects with randomized shape and pose variations; (2) incorporating a mix of diverse objects termed "distractors", positioned in the background near the target objects; and (3) integrating a broad spectrum of random rendering options.

In cryo-ET literature, addressing the synthetic-to-real domain gap for deep learning training has been largely overlooked. Nevertheless, Che et al.[37] proposed a method based on domain randomization for subtomogram segmentation and classification. Building on this, Cryo-Shift[38] utilized domain adaptation and randomization techniques for subtomogram classification, primarily focusing on mapping the synthetic data domain to resemble the real data during classifier training. However, both Che et al.'s work and CryoShift were limited to handling pre-isolated subtomograms of molecules of interest and were not designed to process entire tomograms for particle picking or segmentation.

Here, we introduce "Template Learning", a domain randomization-based strategy for generating cryo-ET simulated tomograms, incorporating considerations for molecular crowding, structural variabilities, and data acquisition variations, with which we can train deep learning models to achieve state-of-the-art performance in particle picking within cryo-ET experimental tomograms. We found that previously unexplored domain randomization axes – the presence of molecular distractors and crowding of the synthetic training data have a crucial impact on the efficacy of target identification in real, entire tomograms. In this paper, we detail our approach and systematically benchmark it against alternative methods, utilizing a recently published, exhaustively annotated in situ cryo-ET dataset. We also apply Template Learning to nucleosome picking within densely populated cryo-ET samples of isolated mitotic chromosomes and compacted interphase chromatin imaged in situ. We illustrate that models trained on simulated datasets, optionally fine-tuned on experimental datasets, outperform those exclusively trained on experimental datasets. Also, we illustrate that Template Learning used as an alternative to template matching can offer higher precision and better orientational isotropy, especially for picking small non-spherical particles.

## Results

### Simulations to train deep learning models on particle picking

Template Learning is a streamlined pipeline for simulating cryo-ET synthetic data for training deep learning models on particle picking. The fundamental goal of Template Learning is to train deep learning models in three aspects:

1. Identifying targeted particles across diverse variations.
2. Differentiating target particles from other structures, especially in crowded environments.
3. Expanding the capabilities of the models to handle the inherent variations present in cryo-ET experimental data.

The practical application of these aspects is achieved through our proposed pipeline, illustrated in Fig. 1 and further elaborated in the subsequent sections.

To enable trained deep learning models to effectively identify various variations of the target biomolecule, we utilize following two strategies:

1. We use multiple templates of one target biomolecule to account for compositional or significant conformational variabilities of the target particles.
2. We generate random flexible variations of each template using Normal Mode Analysis (NMA—a method for fast molecular mechanics simulations, see Methods for details)[39]. The templates with their flexible variations are then simulated at uniformly random orientations to account for orientational variations.

Templates in this study are typically atomic structures. However, since the availability of atomic structures may vary for different target biomolecules, using lower resolution cryo-EM density maps (volumes) is also possible and is explored in a subsequent section.

The existing literature on domain randomization for object recognition in natural images simulates the target objects within a background containing a library of other random objects[40]. These additional objects, distractors, obscure or cast shadows on segments of the target objects. In this study, we selected 100 dissimilar protein assemblies, with diverse molecular weights ranging from 30 kDa to 1 MDa to serve as elements within the background (a comprehensive list of these proteins is available in Supplementary Fig. 1). We inspired our selection of these distractors from the work outlined in TomoTwin, knowing that these structures have different enough feature representations as interpreted by convolutional neural networks allowing to cluster them into corresponding classes. To achieve a similar effect of distractors for cryo-ET data, i.e., overlapping at the level of the tilt series and the missing wedge artifacts, these proteins must be positioned in close proximity to the templates during data modeling.

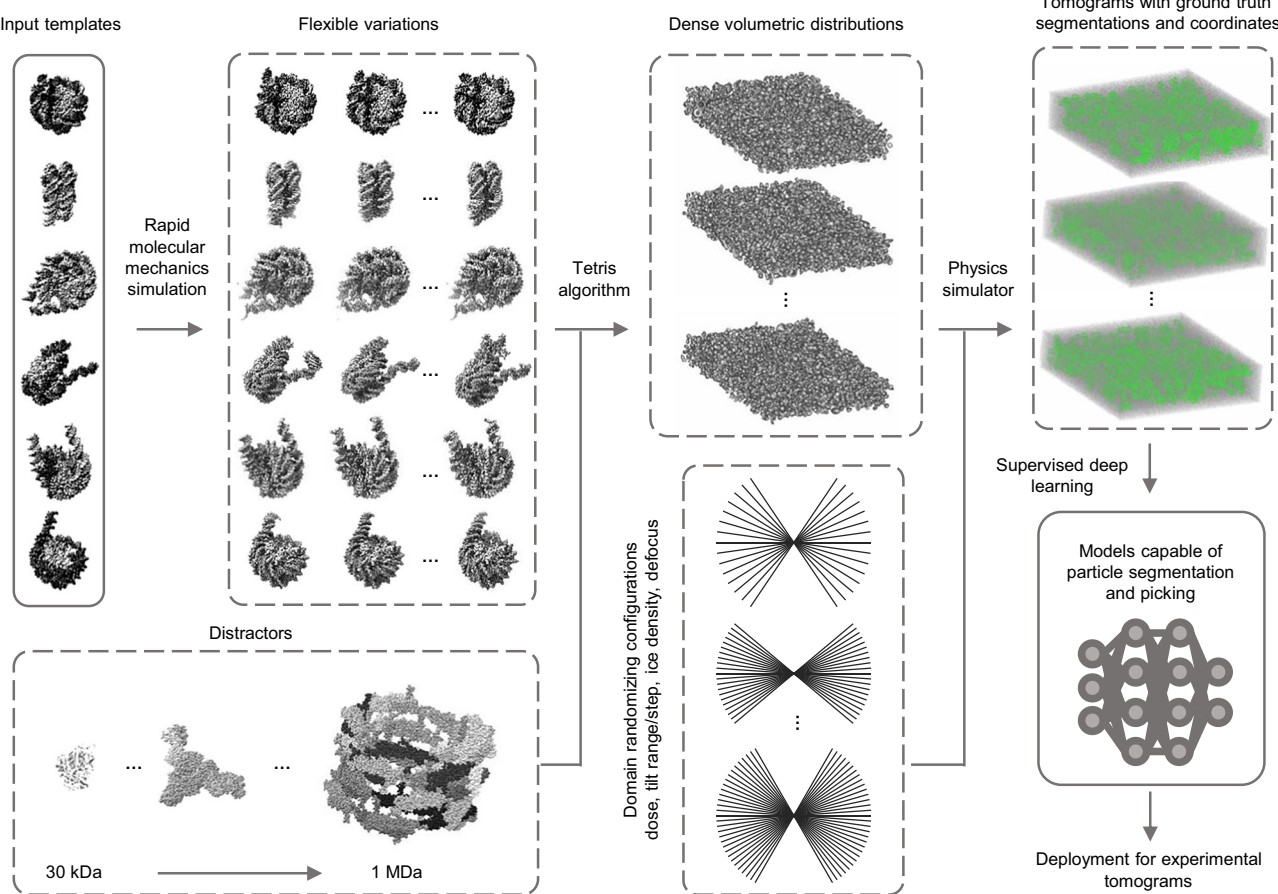

**Fig. 1 | Template Learning workflow for generating simulations used for supervised deep learning used for cryo-ET picking of target particles.** Template structures are used as input, commonly differing by some structural variations of the target biomolecule (we mean by the target biomolecule, the structure that the deep learning model is desired to learn to annotate in cryo-ET tomograms, in the figure, it is the nucleosome). The input templates are augmented with flexible variations based on fast molecular mechanics simulations (see text for details). A general set of other proteins, different from the input templates, are used as "distractors" when generating synthetic data. The templates, along with their flexible variations and distractors are placed at random orientations in close proximity using the "Tetris algorithm"– an algorithm proposed in this work to enable fast simulation of high molecular crowding. A physics-based simulator (Parakeet[28]) is used to generate synthetic cryo-ET data using a set of different parameters that were chosen to produce domain randomization at the lowest cost, which are the electron dose, tilt range, tilt step, ice density, and defocus. The resulting output comprises tomograms, and ground truth segmentations, and coordinates corresponding to the templates. The generated data is subsequently utilized to train deep learning models (e.g., DeepFinder[19]) for experimental tomogram segmentation and particle picking. The PDB IDs used for templates in this figure are 2PYO, 7KBE, 7PEX, 7PEY, 7XZY, and 7Y00, and for distractors are 3QM1, 7NIU, and 6UP6. The display of structures was performed using ChimeraX[54] and IMOD[48] software.

Recent works[27,41] that involve simulating cryo-ET data for practical applications (e.g., deep learning) utilize an iterative brute-force random placement algorithm, involving the rotation of a duplicate of a molecule in each iteration to find a suitable non-overlapping position within the sample. Nonetheless, random placement leads to unstructured empty spaces between the molecules that prevent achieving highly dense samples.

Two other approaches simulate molecular crowding of cryo-ET data[42,43]. The first represents molecules as spheres, based on calculating the minimum bounding sphere for each molecule, and simulates crowding by optimizing a sphere-packing problem. However, this approach is limited to generating high crowding only for spherical-shaped molecules. The second builds on the first by including an additional step of molecular dynamics simulations to enhance packing density. However, routine application of this approach requires significant computational resources.

To achieve high molecular crowding while maintaining generality and efficiency, we drew inspiration for a simplified approach from a recent concept on packing generic 3D objects[44], a solution we term the "Tetris algorithm." Our algorithm, illustrated in Fig. 2, operates on an intuitive principle: it places molecules iteratively, with each iteration positioning a new molecule at a uniform random orientation as close as possible to those already placed (the user only needs to choose the minimum distance). For more details on the principles of the Tetris algorithm, see "Methods".

Simulations can partially replicate the domain of real-world data; however, variations in the real world can be unpredictable, and simulations simplify some complex phenomena for practical reasons. Thanks to domain randomization[34,36], simulating an exact match of the real world is not necessary for training deep learning models capable of generalizing to real-world data. Instead, domain randomization focuses on training these models on diverse simulated scenarios to minimize the influence of the domain on their capabilities.

In this work, we employ a cryo-ET physics-based simulator called Parakeet[28]. Parakeet, and its MULTEM backend[45], provide control over numerous parameters for simulating cryo-ET hardware and the behavior of biological samples during data recording. While many parameters related to the electron beam, lenses, detector, sample, and data acquisition strategy can be controlled, it might not be necessary to randomize every available parameter. On one hand, the combinations

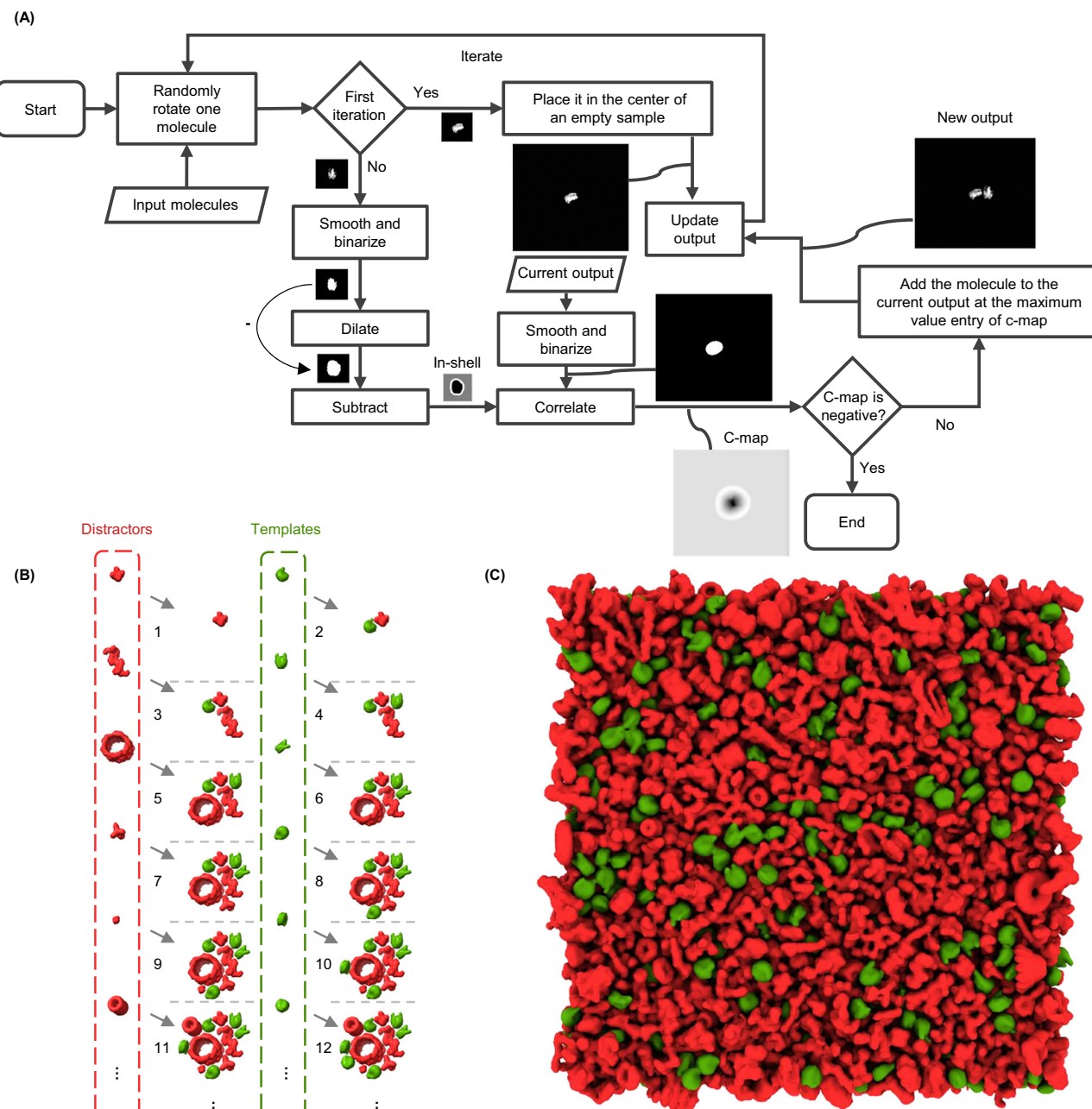

**Fig. 2 | The Tetris algorithm for generating crowded 3D samples. A** The flowchart of the algorithm, demonstrated in 2D for simplicity. The molecules that are desired to populate the sample are used as input. In the initial iteration, a molecule is positioned at the center of an expanded volumetric sample, forming the current output. In subsequent iterations, a new molecule undergoes binarization and dilation. By multiplying the binarized version with a large positive value and subtracting the result from the dilated version, an insertion shell is generated (referred to as 'in-shell'). This insertion shell is then correlated with a binarized version of the current output volume, resulting in a correlation map (referred to as 'c-map'). Positive values in the c-map (white voxels) represent viable positions for adding the new molecule without intersection, whereas negative and zero values (black and gray voxels, respectively) represent positions where intersection with existing molecules occurs or where the distance from other molecules is too large. If the correlation map is not entirely negative, the molecule is added to the current output at the index of the maximum entry, ensuring it avoids intersections while maintaining the highest compactness. **B** An example of the Tetris algorithm input and output at each iteration, alternating between placing templates (in green) and distractors (in red). The numbers in (**B**) correspond to the output at different iterations of the algorithm in (**A**). **C** An example of a Tetris algorithm output in 3D.

required to cover all conditions grow exponentially with the number of variables. On the other hand, some variables might have undesirable effects on the overall simulation. Here, we propose varying a few essential parameters that significantly influence the output simulation: the electron dose, defocus, tilting range, tilting step, and ice density, while giving other parameters default values. The electron dose is crucial for varying the SNR and the irradiation effects on the sample.

The defocus plays a key role in controlling the Contrast Transfer Function (CTF)[46]. Varying the tilting range and step may not replicate experimental data acquisition strategies, however, it can expose the deep learning model to variations that increase its robustness, for example, to artifacts and imperfections resulting from tilt series alignment and reconstruction. Lastly, different ice densities can simulate the variability of sample thickness and solvent composition. A

combination of three values for the defocus and two values for each of the other variables results in 48 combinations that we found sufficient for efficient domain randomization.

## Benchmarking Template Learning for picking ribosomes in situ

In a recent study[20], the first exhaustively annotated in situ cryo-ET dataset was published (EMPIAR-10988). Here, we use this dataset to evaluate the proposed Template Learning thoroughly.

In the following, we generate, based on the Template Learning workflow, a simulated dataset to train DeepFinder[19] for ribosome annotation. We compare the performance of this model (i.e., Deep-Finder trained solely on simulations) to the contemporary techniques, namely, DeePiCt[20] and DeepFinder (trained solely on annotated experimental data), and 3D template matching (we will refer to it as template matching hereafter).

We organized our experiments based on two principles. First, we benchmark the complete method that integrates the aforementioned concepts of Template Learning. Then, we conducted experiments where specific elements from the data simulation pipeline were intentionally omitted (e.g., the structural variations of the target template, the crowding, etc.) to assess their impact on performance and to offer guidelines for future users. Second, acknowledging that the size of a simulated dataset is user-defined (generally in deep learning, the more training data, and the more diversity, the better), we ensured the feasibility of routine use of the method by limiting its time requirement to a maximum of two days on a single GPU (within one day on a typical computing node with 4 GPUs, see Code Availability for details).

To establish a Template Learning workflow, we proceeded with the following steps. We selected 6 eukaryotic ribosome PDB structures (PDB IDs: 4UG0, 4V6X, 6Y0G, 6Y2L, 6Z6M and 7CPU).

The 6 selected structures differ slightly in composition or conformation. By applying NMA, we generated 25 flexible variations for each structure (refer to the NMA section in Methods for details). While generating synthetic data, we employed the aforementioned general distractors (a full list of distractors is given in Supplementary Fig. 1). To simulate a crowded environment, we generated volumes from the template structures (i.e., the 6 ribosome PDBs and their flexible variations) and distractors with a sampling rate of 16 Å using Eman2[47] software e2pdb2mrc. We used the generated volumes as input to populate 48 crowded volumetric distributions of size $192 \times 192 \times 64$ voxel[3] using the Tetris algorithm, with ribosome templates appearing at a frequency of one for every 5 distractors to produce a balanced volumetric density ratio balance between distractors and templates in the simulated tomograms (see Tetris algorithm section in Methods for details). Subsequently, we fed the sample information (i.e., atomic structures of templates and distractors with the positions and orientations of crowding generated using the Tetris algorithm) to Parakeet[28] to simulate 48 tilt series using a dose symmetric tilting scheme using the parameters listed in Supplementary Table 1. All other parameters adhered to default values for Volta Phase Plates (VPP) simulations within the Parakeet software. Subsequently, we binned the simulated tilt series to a similar sampling rate as the analyzed dataset (13 Å/pixel) and reconstructed them in IMOD[48] using Weighted Back Projection. We used the tomograms and their corresponding ground-truth template coordinates (around 6500 simulated ribosomes) and segmentations (i.e., volumes of the same size as the tomograms where template instances appear in white over black background, illustrated in Supplementary Fig. 2) to train DeepFinder[19] using its default parameter settings (the list of parameters is given in Supplementary Table 2).

We applied the DeepFinder simulations-trained model to the 10 VPP tomograms of the EMPIAR-10988 dataset without any preprocessing, resulting in segmentation maps, i.e., ribosome segmentations based on the decisions of the model. Subsequently, we employed the MeanShift algorithm offered by the DeepFinder software, using a clustering radius of 10 voxels, to extract both the coordinates and size of these segments (count of the number of voxels of each segment), with and without the application of a mask (specifically, a cytosol mask sourced from the dataset, utilized to eliminate false positives outside the mask; see Supplementary Note 1 for more details about these masks). We compare the extracted coordinates to the expert-validated annotations provided by the dataset at different levels of segment sizes used as a threshold (i.e., segments smaller than the threshold were removed). For this comparison, we maintained the criterion that two annotations—one from the output of the deep learning model and the other from the expert annotations—were considered to target the same particle if their spatial distance was within 10 voxels (a value consistent with the methodology of a previous study analyzed the same dataset[20]). An example of a tomogram, its corresponding segmentation map, and the coordinates set at a threshold that removed smaller objects compared to expert annotations are presented in Fig. 3. To evaluate the performance of the method, we computed the overall Recall and Precision with and without masking computed over all the tomograms jointly (refer to the Methods section for more details on the assessment metrics). The point where Precision and Recall curves intersect was utilized to determine the $F_1$ score per tomogram for the dataset (shown in Fig. 4A). The results, further illustrated in Fig. 4I, J, reveal that Template Learning used to train Deep-Finder exclusively on simulations, achieved state-of-the-art performance, outperforming template matching and previous deep learning methods trained solely on annotations derived from the same experimental dataset based on the median $F_1$ score (compared to values reported in ref. 20).

To reflect on these results, it is essential to highlight that the previously reported findings for DeePiCt and DeepFinder were based on models trained with 8 out of 10 fully annotated tomograms, containing approximately 20,000 ribosome annotations[20]. Hence, previous evaluations were conducted on 2 out of 10 tomograms, for annotating the remaining 5000 ribosomes, utilizing three cross-validation splits (in each split, a random set of 8 tomograms was used for training and the remaining 2 tomograms for benchmarking). Additionally, each of the two models underwent a different training strategy to achieve optimal performance. In particular, in the case of DeepFinder, the training encompassed two classes—ribosomes and Fatty Acid Synthase (FAS)—as training solely on ribosomes led to suboptimal performance. On the other hand, DeePiCt was exclusively trained on ribosomes, as unlike DeepFinder, combining ribosomes and FAS during training led to suboptimal performance.

In contrast, the DeepFinder model trained solely on the simulations generated from the Template Learning workflow, not fine-tuned on any annotated experimental data, was benchmarked on the complete dataset (i.e., all 10 tomograms). Hence, despite the relatively modest increase in the $F_1$ score for the Template Learning method (0.85, compared to the previous best of 0.83), it stands as a demonstration that deep learning can be trained effectively for picking a target structure, starting from only prior templates, and domain-randomized simulations. Also, a model of the same network architecture (DeepFinder) trained only on simulations outperforms its counterpart trained only on annotated experimental datasets for cryo-ET particle picking.

To systematically assess the contribution of each component of the Template Learning workflow, we performed a series of ablation and variation experiments, as detailed in the Supplementary Materials - Template Learning variations and Supplementary Notes 3 and 4. These analyses revealed that both incorporating multiple atomic structures and generating flexible molecular variations enhance annotation performance, and that introducing artificial flexible variations can partially compensate for the absence of multiple templates (Fig. 4A–C compared to Fig. 4D and Supplementary Fig. 3). The

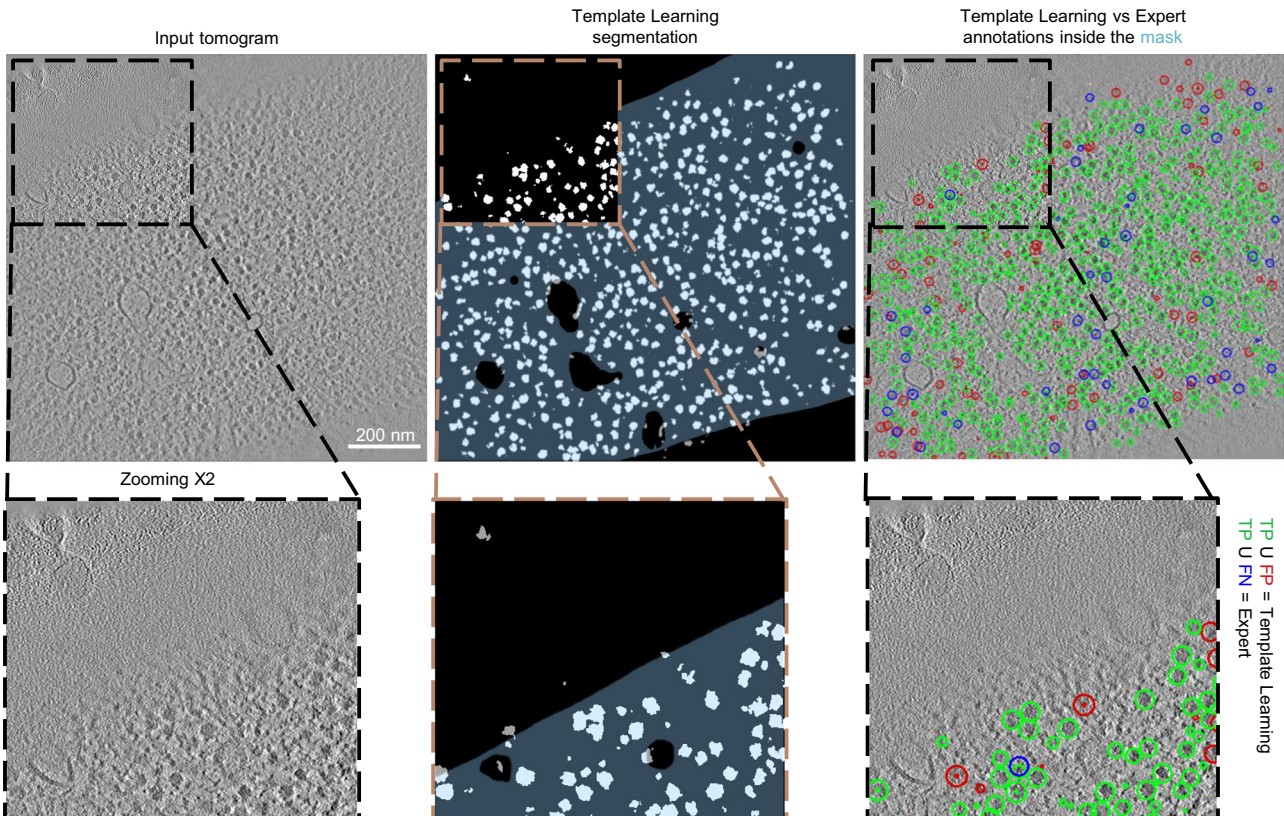

**Fig. 3 | Template Learning trains DeepFinder to single molecule precision on segmenting and annotating ribosomes in situ.** Left: Display of a 2D central slice of a VPP tomogram. Middle: Segmentation of ribosome from the Template Learning workflow shown in black and white, with a 50% transparent overlay of the mask. Right: resulting annotations within the mask, obtained by converting segmentations to annotations using the MeanShift algorithm (provided by the DeepFinder[19] software) using a clustering radius of 10 voxels, compared to expert annotations. Comparison includes True Positives (TP), False Positives (FP), and False Negatives (FN) relative to expert annotations. The tomogram, mask and expert annotations are sourced from the EMPIAR-10988 dataset.

comparative impact of these design choices is illustrated by recall, precision, and $F_1$ score. The inclusion of a diverse set of distractor molecules was found to be critical for high precision in particle picking; omitting or restricting distractor variability substantially increased false positive rates (Fig. 4E, F). Furthermore, simulating densely crowded environments via the Tetris algorithm (Supplementary Fig. 4) was essential, as reducing crowding in training simulations led to a marked drop in median $F_1$ score (Fig. 4G). We also validated the use of low-resolution volumetric templates by developing a real-time conversion to pseudoatomic models compatible with physics-based simulations; this approach enabled training with reasonable recall, albeit at the expense of some precision (Fig. 4H). In addition, Template Learning could be readily adapted to different imaging domains, such as VPP and conventional defocus (DEF), and offered improved performance on cross-domain tasks compared to previous approaches (Fig. 5 and Supplementary Fig. 5).

**Fine-tuning Template Learning-trained models using real data**

The literature on domain randomization indicates that models pretrained on simulations and subsequently fine-tuned on real-world data outperform models trained exclusively on real-world data[35]. This approach can be particularly valuable to cryo-ET particle picking studies that involve challenging-to-annotate molecules, or when training on domain randomization simulations alone does not yield satisfactory results.

In the same dataset used for benchmarking Template Learning on ribosomes in situ (EMPIAR-10988), another molecule, FAS, is annotated. Previous attempts based on template matching and supervised

deep learning on experimental datasets have reported challenges in annotating FAS, attributed to its distinctive shell-like structure, where particles have lower SNR compared to ribosomes[20].

In the following, we present the benchmarking of Template Learning combined with DeepFinder, initially trained exclusively on a simulated dataset, on FAS annotation in EMPIAR-10988 tomograms. Subsequently, we gradually introduce fine-tuning based on the experimental data annotations to assess whether performance can be improved.

To establish a Template Learning workflow for FAS, we utilized two PDB structures (PDB IDs: 4V59 and 6QL5). By using NMA, we generated 80 flexible variations for each structure (refer to the NMA section in the Methods for details). We kept the remaining parameters for simulated data generation consistent with those aforementioned for the ribosome study, except for training the models on sphere segmentations due to the shell-like structure of FAS.

The $F_1$ score results of the simulation-trained DeepFinder model for FAS annotations are presented in Fig. 7. The results show that on the VPP dataset, the DeepFinder model trained on the Template Learning workflow outperformed that trained exclusively on experimental data. The results on the DEF dataset gave an $F_1$ score ranging up to 22%, outperforming previous attempts that failed to pick any particle, based on the results reported in ref. 20. However, these results can still benefit from improvements compared to previously obtained results with DeePiCt trained on annotated experimental data.

We proceeded to fine-tune the simulations-trained DeepFinder model progressively using annotations from experimental data. In the following experiment, we fine-tuned it using annotations from 2 VPP

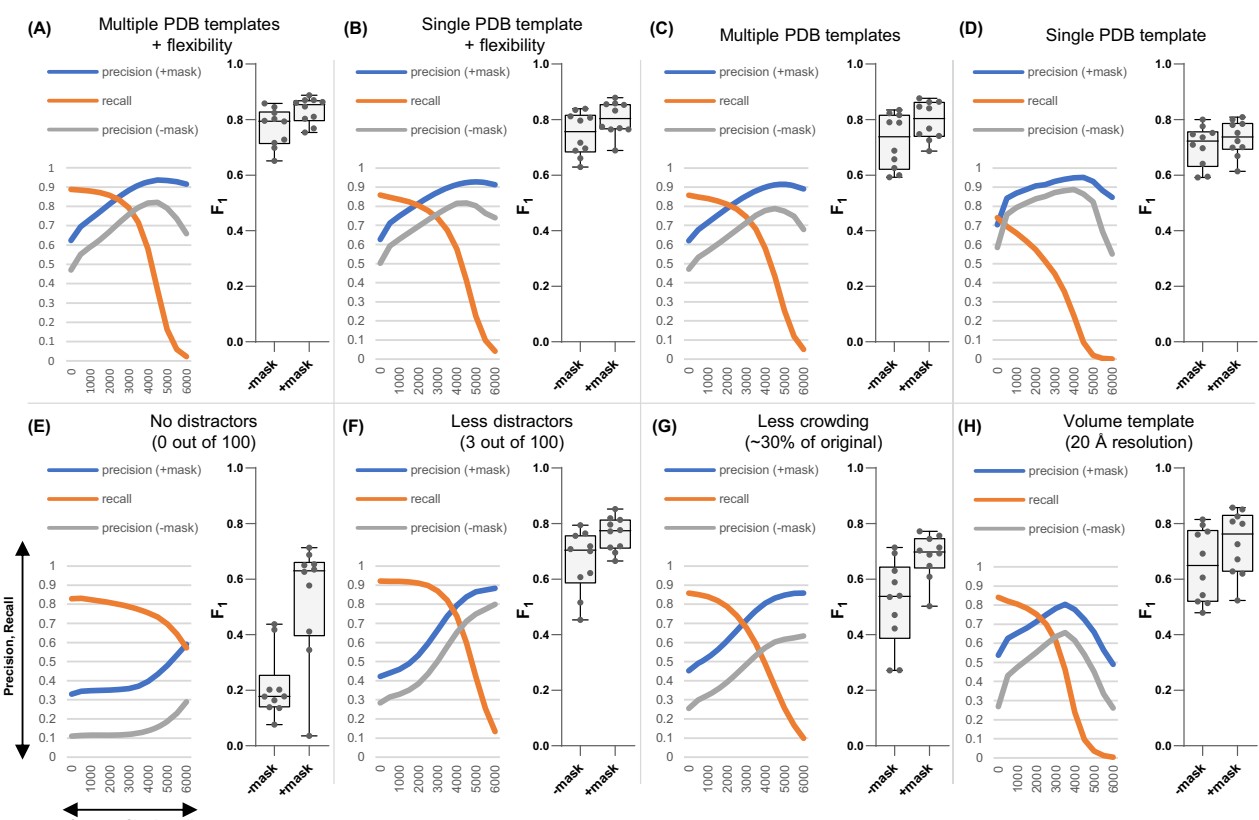

**(I)                                    Template Learning variations (A is the full method)**

| Deep learning models (DeepFinder) trained solely on simulations – benchmarked on 10/10 experimental tomograms (25k annotations) | | | |
|---|---|---|---|
| Principle studied | Distinction of the simulated dataset | Without mask $F_1$ median | With cytosol mask $F_1$ median |
| Incorporating multiple atomic template structures and flexible variations | **A**  Multiple PDB templates + flexibility | 0.79 | **0.85** |
| | **B**  Single PDB template + flexibility | 0.76 | 0.80 |
| | **C**  Multiple PDB templates | 0.74 | 0.80 |
| | **D**  Single PDB template | 0.72 | 0.74 |
| Employing distractors | **E**  No distractors (0 out of 100) | 0.18 | 0.63 |
| | **F**  Less distractors (3 out of 100) | 0.70 | 0.77 |
| Simulating crowding | **G**  Less crowding (~30% of original) | 0.54 | 0.70 |
| Using a volumetric template | **H**  Volume template (20 Å resolution) | 0.65 | 0.76 |

**(J)                                                    Previous results**

| Method | Training / Benchmarking | With cytosol mask $F_1$ median |
|---|---|---|
| DeePiCt (trained solely on experimental data) | 8 tomograms (20k annotations) / 2 tomograms (5k annotations) | 0.79* |
| DeepFinder (trained solely on experimental data) | | 0.83* |
| Template Matching | NA / 10 tomograms (25k annotations) | 0.49 |

*median is reported for 3 cross-validation experiments with random splits

tomograms containing approximately 150 FAS annotations. We kept all the DeepFinder training parameters to default (listed in Supplementary Table 2), except for reducing the number of steps and epochs to 10 steps for 10 epochs (in place of 100 steps for 100 epochs) to avoid overfitting (since the number of training data examples is low). We performed three cross-validation experiments (i.e., in each experiment, the data is randomly split into 2 tomograms for training and 8 for benchmarking). The results presented in Fig. 6 show a significant increase in the median $F_1$ score compared to training on simulated data

only. Notably, de Teresa-Trueba et al.[20] previously reported that training a deep learning model with fewer than 600 particles was ineffective. Therefore, the ability to fine-tune our model with only 150 particles after pre-training on simulations represents a major advancement over previous supervised methods that required larger datasets to train models from scratch using only experimental annotations.

In a subsequent experiment, we fine-tuned the simulation-trained DeepFinder model on 8 VPP tomograms containing approximately

**Fig. 4 | DeepFinder trained on Template Learning simulations only outperforms previous techniques for ribosome annotations in cryo-electron tomograms.** Performance benchmarking and comparative analysis of the Template Learning on ribosome annotation on the S. pombe dataset (EMPIAR-10988, 10 VPP tomograms). **A–H** Performance measures for 8 variations of Template Learning settings: performance in all these experiments was evaluated on $n = 10$ independent tomograms from S. pombe in a single experiment, acknowledging that models trained solely on simulated data yield deterministic results on experimental data, making repeated identical evaluations unnecessary. The curves depict the overall Precision and Recall against the volume of the segmented region (horizontal axis). Precision is evaluated with and without masking (± mask). Boxplots show the $F_1$ score per tomogram at the threshold where the Precision and Recall curves intersect, which often coincides with the highest overall $F_1$ score (except for the curve shown in E (− mask), where the threshold was chosen based on the intersection

point from the (+ mask) case). Box plot middle lines mark the mean value, and the edges indicate the 25th and 75th percentiles; whiskers indicate the range of existing data (no outliers were removed). **I** Table listing the differences between the experiments in (**A–H**), comparing their results based on their median $F_1$ scores (bold value is the highest among proposed and previous techniques). **J** Table of previous results with their reported performance based on the findings in ref. 20. Table entries marked with an asterisk (*) indicate when values reported as the median of 3 independent cross-validation experiments, where each experiment used a random split of $n = 8$ tomograms for training and $n = 2$ tomograms for validation (out of the 10 original tomograms). Comparison of segmentations resulting from the different experiments on an example tomogram are shown in Supplementary Fig. 3. Descriptive statistics and statistical tests validating the significance of these Template Learning variations are provided in Supplementary Note 2.

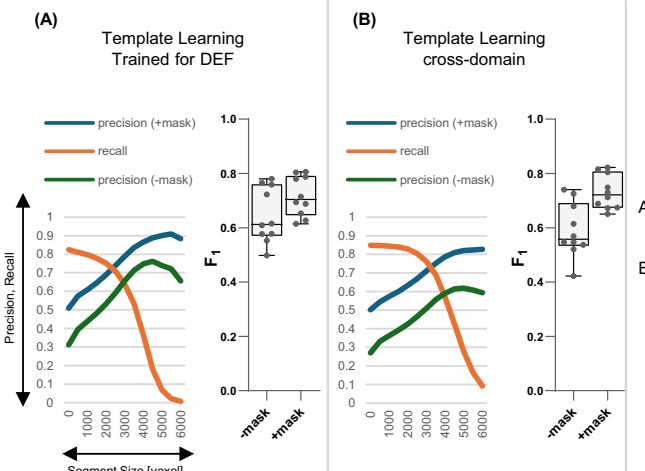

**Fig. 5 | Template Learning does not necessitate data preprocessing for DEF tomograms.** Performance benchmarking and comparative analysis of Template Learning workflow applied to ribosome annotations within the S. pombe dataset (EMPIAR-10988, 10 DEF tomograms). **A**, **B** The curves depict the overall Precision and Recall against the volume of the segmented region (horizontal axis). Precision is evaluated with and without cytosol masking (±mask). Boxplots show the $F_1$ score per tomogram at the threshold where the Precision and Recall curves intersect, which coincides with the highest overall $F_1$ score. Performance in all these

experiments was evaluated on $n = 10$ independent tomograms from S. pombe in a single experiment, acknowledging that models trained solely on simulated data yield deterministic results on experimental data, making repeated identical evaluations unnecessary. Box plot middle lines mark the mean value and the edges indicate the 25th and 75th percentiles; whiskers indicate the range of existing data (no outliers were removed). **C** On the top, a table listing the differences between the experiments in (**A**, **B**), comparing their results based on their median $F_1$ scores. On the bottom, previously reported performance based on the findings in ref. 20.

600 FAS annotations and inferred the model on the remaining 2 tomograms, using 3 cross-validation splits. Again, we kept all the DeepFinder training parameters to default, except for the number of steps and epochs, keeping 10 steps for 30 epochs (again, to prevent overfitting). The results presented in Fig. 6 show a significant increase in the median $F_1$ score, outperforming previous supervised deep learning methods trained solely on experimental data.

Finally, we performed a cross-domain experiment, where we finetuned the simulation-trained DeepFinder model on the VPP dataset and applied this model to annotate FAS in the DEF dataset after SM preprocessing. Consistent with previous findings, the results presented in Fig. 6 show that training on simulations and fine-tuning on experimental data outperforms training only on experimental data.

### Template Learning improves precision and isotropy in nucleosome picking

In cryo-ET data processing, localizing a target biomolecule in a new dataset is a common objective. The application of supervised deep learning to this task requires an initial training dataset of particles of interest extracted directly from the new data. Such a dataset is usually in the order of thousands of particles in different orientations, where more particles are needed for better performance. Manually

annotating these particles is time-consuming, making template matching followed by manual elimination of obvious false positives and further curation through STA and classification the most common approach[19,20]. The principle of Template Learning offers a time and computing-efficient alternative to this complex procedure by annotating new data directly in a single step using a model trained on synthetic data.

In this section, we explore the efficiency of Template Learning for nucleosome annotation within a new cryo-ET dataset of partially decondensed mitotic chromosomes in vitro (refer to the Method section for sample preparation details) and compare it with a template matching-based annotation routine.

A tomogram central slice of our data is shown in Fig. 7A. Despite this dataset being isolated chromosomes in vitro, it contains an abundance of structures other than nucleosomes that can be false positives which are: DNA linkers, gold nanoparticles, percoll, and the non-histone components of chromatin.

To establish a Template Learning workflow, we utilized six nucleosome template structures with PDB IDs: 2PYO, 7KBE, 7PEX, 7PEY, 7XZY, and 7Y00.

By applying NMA, we generated 100 flexible variations for each structure (more details can be found in the NMA section of "Methods").

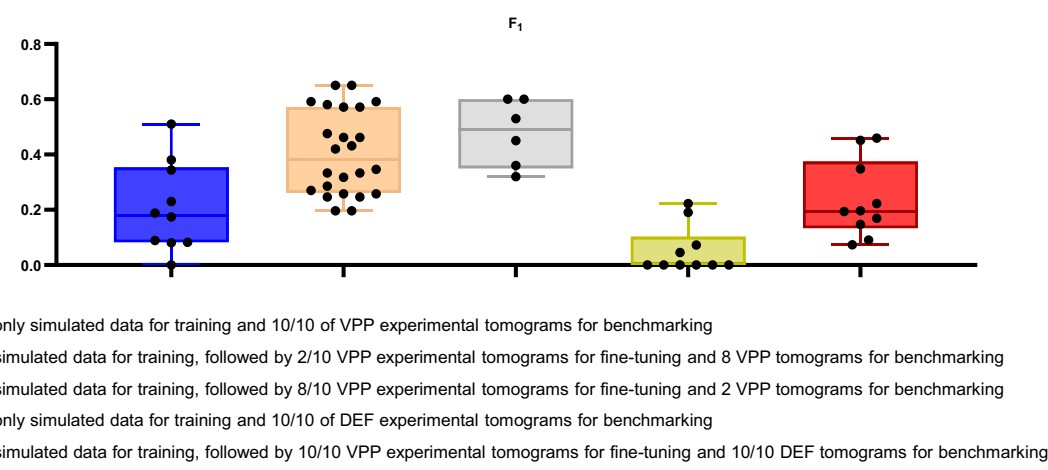

Benchmarked on DEF experimental tomograms

|  |  | Method | $F_1$ median |
|---|---|---|---|
| Template Learning | 🟨 | Using only simulated data for training and 10/10 of DEF experimental tomograms for benchmarking | 0 |
|  | 🟥 | Using simulated data for training, followed by 10/10 VPP experimental tomograms for fine-tuning and 10/10 DEF tomograms for benchmarking | 0.20 |
| DeePiCt |  | Using 10/10 VPP experimental tomograms for training and 10/10 DEF tomograms for benchmarking | 0.15 |

Benchmarked on VPP experimental tomograms

|  |  | Method | $F_1$ median |
|---|---|---|---|
| Template Learning | 🟦 | Using only simulated data for training and 10/10 of VPP experimental tomograms for benchmarking | 0.18 |
|  | 🟧 | Using simulated data for training, followed by 2/10 VPP experimental tomograms for fine-tuning and 8 VPP tomograms for benchmarking | 0.38* |
|  | ⬜ | Using simulated data for training, followed by 8/10 VPP experimental tomograms for fine-tuning and 2 VPP tomograms for benchmarking | 0.49* |
| DeePiCt |  | Using 8/10 VPP experimental tomograms for training and 2 VPP tomograms for benchmarking | 0.46* |
| DeepFinder |  | Using 8/10 VPP experimental tomograms for training and 2 VPP tomograms for benchmarking | 0.11* |

*median is reported for 3 cross-validation experiments with random splits

**Fig. 6 | DeepFinder can be pre-trained on Template Learning simulations and fine-tuned on experimental data annotations.** Performance benchmarking and comparative analysis of Template Learning applied to FAS annotation within the S. pombe dataset (EMPIAR-10988, 10 DEF tomograms). The results show the performance of the Template Learning workflow used for training DeepFinder on simulations, followed by different levels of fine-tuning on experimental data, compared to results based on the findings in ref. 20. Table entries marked with an asterisk (*) indicate when values reported as the median of 3 independent cross-validation experiments, where each experiment used a random split of either $n = 2$ or 8 tomograms for training and $n = 8$ or 2 tomograms for validation (out of the 10 original tomograms, indicated in the table). Otherwise (for entries without an asterisk), performance was assessed on all $n = 10$ tomograms in a single experiment, acknowledging that models trained solely on simulated data yield deterministic outcomes on experimental data, making repeated identical evaluations unnecessary. Box plot middle lines mark the mean value, and the edges indicate the 25th and 75th percentiles; whiskers indicate the range of existing data (no outliers were removed).

Notably, these templates represent different compositional variations of the nucleosome. The 7PEX structure includes the H1 protein, while the other five templates lack H1 but exhibit varying DNA linker lengths.

The remaining parameters of the Template Learning workflow were configured similarly to those used for the workflow established for ribosome and FAS annotation, with a few notable exceptions explained below.

Firstly, recognizing the smaller size of nucleosomes compared to ribosomes, we adjusted the frequency of appearance of nucleosome templates to one nucleosome for every two distractors (in contrast to one ribosome for every five distractors). This adjustment aimed to maintain a balanced volumetric density ratio between distractors and templates in the simulated tomograms.

Secondly, recognizing that nucleosomes have fewer atoms than ribosomes, we observed increased speed in the execution of the physics-based simulator (i.e., Parakeet). This allowed for the generation of larger tomograms for training, all within the same runtime for the ribosome study (see Code Availability for details). In particular, the tomogram size generated in the Tetris algorithm here was set to

256 x 256 x 64 (for a pixel size of 16 Å), in contrast to 192 x 192 x 64 used for ribosome simulations.

Lastly, we adjusted the pixel size of the simulated data to 8 Å through binning, closely approximating the pixel size of our experimental tomogram, which had been previously binned four times before annotation (the unbinned pixel size was 2.075 Å).

Consequently, we trained DeepFinder on the resulting Template Learning simulations and applied it to segment our data. The corresponding segmentation map (score map), is shown in Fig. 7B. Upon visual inspection, it is evident that that model has high Precision in localizing nucleosomes. Subsequently, we employed the MeanShift algorithm from the DeepFinder software to extract the corresponding annotations for the nucleosomes, utilizing a clustering radius of 6 pixels, approximately equivalent to the nucleosome's radius at this pixel size. We excluded particles close to the edges of the tomogram, the air-water interface, and the sample-carbon interface.

Following this process, we obtained 18 k annotated particles and proceeded with the reference-free STA procedure using Relion V4.0[15] by generating an initial model from the data, followed by two stages of

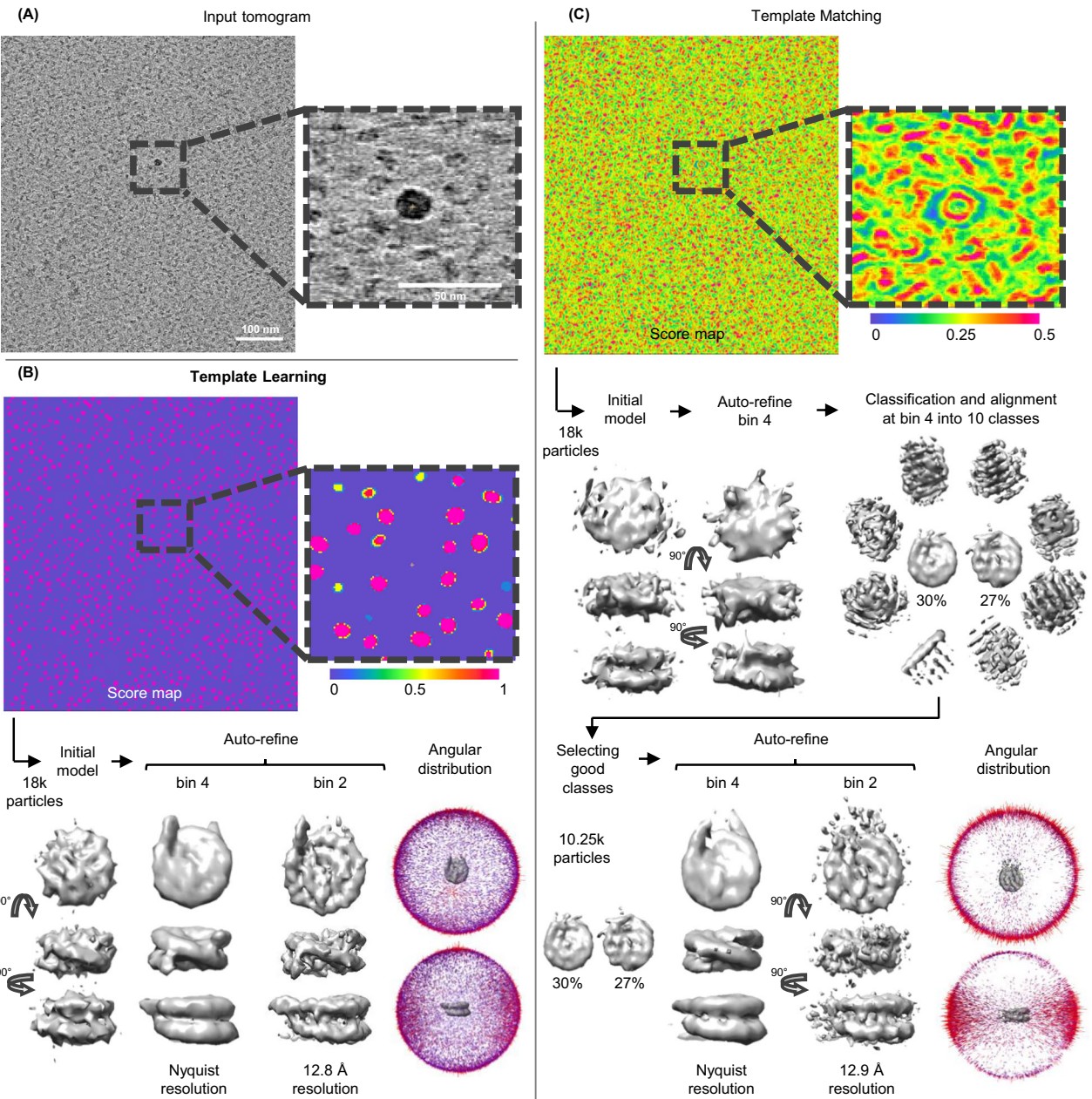

**Fig. 7 | Template Learning outperforms traditional template matching in Precision and orientational isotropy in annotating nucleosomes. A** Central slice and zoomed-in 10-slice average of a cryo-ET tomogram of partially decondensed mitotic chromosomes in vitro. **B**, **C** The results of applying Template Learning (6 PDB structures as input, other parameters kept default) and traditional template matching (using PyTom with a nucleosome template) for annotating nucleosomes within the tomogram in (**A**), followed by reference-free STA in Relion V4. The experiments were repeated on 4 independent tomograms (N = 4) with similar results. **B** Top: score map resulting from training DeepFinder on nucleosome annotations using simulated datasets only from the proposed Template Learning workflow. Below: results of STA using 18 k annotations extracted using a 6 voxel clustering radius of the MeanShift algorithm (provided by the DeepFinder software). The results show initial model generation, followed by two stages of auto-refinement, and the angular distribution of averaged particles. The refinement has converged without particle curation, and the angular distribution of averaged particles is isotropic. **C** Top: score map resulting from traditional template matching. Below: results of STA using the top 18 k local maxima from the score map. The first step was an initial model generation, followed by a stage of refinement. The first stage of refinement did not improve the resolution of the initial model, therefore, it was followed by simultaneous classification and alignment into 10 classes. The particles corresponding to nucleosome-like classes (57% of the original 18 k particles) were used for two stages of refinement. The results show a slightly lower resolution average than the one obtained using Template Learning and show an anisotropic angular distribution.

refinement at binning 4 and binning 2, resulting in resolving the nucleosome structure at 12.8 Å (at 0.143 FSC threshold) resolution shown in Fig. 7B.

The angular distribution of the averaged particles seems isotropic, showcasing that the Template Learning method is not biased to certain orientations.

To further verify the Precision of nucleosome identification, we classified the particles into 10 and 50 classes (presented in Supplementary Fig. 6). All class average results showed well-recognizable nucleosome shapes with DNA gyres, with the major differences being in the heterogeneity of the DNA entry-exit segments. We also performed a local resolution assessment using ResMap[49] (presented in

Supplementary Fig. 7) and observed that resolution has ranged from 12–20 Å, with the highest resolution in the DNA close to the core histones, and the lowest resolution around the DNA entry-exit segments.

Our averaging and classification results show that our Template Learning workflow generated precise nucleosome annotations of uniform orientations, facilitating high-throughput STA without the need for multiple rounds of classification to eliminate false-positive particles.

We applied template matching starting from a nucleosome template structure generated from the PDB 2PYO using the default procedure in PyTom[18]. We then applied the template matching procedure to our new data using an angular sampling of 7° increments (45,123 rotations).

The score map is shown in Fig. 7C. The visual inspection of the score map shows some high signal for some nucleosomes also identified by the Template Learning procedure (Fig. 7A), but also obvious false positive signal for other objects (an example of high false signal for the Percoll particles is shown at the center of the zoomed-in image in Fig. 7C).

To have a meaningful and fair comparison with the Template Learning procedure, we extracted from the same tomographic region the 18 k particles showing the highest cross-correlation scores. Unlike the results of Template Learning, a brief visual inspection of the template matching peaks showed 50 obvious false positives corresponding to the 10 nm gold and Percoll particles. The false positives were removed, and the data were processed Relion V4.0[15]. The reference-free initial model was generated, followed by a stage of refinement at binning 4. Unlike Template Learning, the refinement did not result in a significant improvement of the resolution of the initial model (judged by the shape and the FSC), indicating the presence of a significant ratio of non-nucleosome particles among the annotations. In agreement with this assumption, classification, and alignment into 10 classes at the same binning resulted in only 2 nucleosome-like classes that sum to 10.25 k picks (57% of the original picks). The particles of these 2 classes were selected for a further refinement process at binning 4 reached Nyquist resolution, and the further refinement at binning 2 led to 12.9 Å resolution, which is comparable to that achieved with Template Learning.

Importantly, the angular distribution of nucleosomes annotated by template matching showed a significant imbalance towards side views compared to round top views (Fig. 7C). Our simulations (Supplementary Fig. 8) showed that due to the cylindrical shape of nucleosomes and the missing wedge problem of cryo-ET data, the constrained cross-correlation between a template and the particles is a function of the orientation, where side views show higher cross-correlation than top and oblique views. This problem results in extracting only side views of nucleosomes at an adequate Precision.

The orientation bias of template matching hampers its efficiency for particle detection and is prone to resolution loss in STA. In our case, the resolution of the nucleosome was not affected significantly, because its pseudo-symmetric cylindrical shape allowed the complete 360° orientation range of side views to have cross-correlation sufficient for detection (Fig. 7C). However, this constraint may pose a more significant challenge when annotating particles with asymmetric non-spherical shapes. To further validate the Template Learning pipeline in a crowded cellular environment, we compared its performance with supervised deep learning and template matching for nucleosome annotation within a tomogram of a vitreous section from Drosophila embryonic brain tissue[8,50,51]. Our STA results demonstrated that Template Learning outperformed the other two approaches in terms of precision and orientational isotropy (see Supplementary Note 5). Notably, template matching, a manual expert annotation, and a supervised model trained on this annotation exhibited significant orientational bias, favoring nucleosome side views. Our results

demonstrate that Template Learning is capable of overcoming this limitation.

## Discussion

In this study, we introduced Template Learning, an approach that blends the simplicity of Template Matching, requiring only template structures as input, with the advanced capabilities of Deep Learning, thereby reducing the dependence on extensive, labor-intensive annotated datasets for supervised training. While previous works[37,38] applied domain randomization by simulating acquisition information of macromolecules simulated as subtomograms in isolation, our method effectively tackles key challenges in cryo-ET particle picking, such as structural variability, cellular crowdedness, and experimental data variance, through a refined implementation of domain randomization to create simulated data for deep learning training.

The efficacy of Template Learning arises from the implementation of new domain randomization axes - distractors and crowding, together with structural variability and physics-based simulation. These aspects enable the well-established DeepFinder model, known for its particle annotation capabilities, to efficiently train without the need for highly realistic cellular environment simulations.

This approach simplifies the process for users, allowing them to generate effective simulations that train DeepFinder in annotating specific biomolecules in cryo-ET data, using only templates in a streamlined yet customizable simulation process. Template Learning can utilize any atomic models, including those generated by artificial intelligence-based tools, as templates. We also confirm the feasibility of relevant template generation from low-resolution cryo-EM densities.

In this paper, we employed a comprehensively annotated in situ cryo-ET dataset for ribosomes and FAS, covering defocused cryo-ET and VPP, to validate Template Learning's efficacy and versatility. We demonstrated how to enhance DeepFinder's supervised training by initially training on simulations and then fine-tuning on experimental data. Notably, DeepFinder, when trained solely on simulations, surpassed previously established training on experimental annotations for ribosomes. Furthermore, the combined approach of pre-training on simulations and subsequent fine-tuning on experimental data showed improved performance for FAS. Nonetheless, fine-tuning remains heavily dependent on the quality of training examples and their representativeness of new data, a general limitation in supervised learning.

We evaluated the efficacy of our method for the identification of a known target molecule in tomographic data without any prior annotation, focusing on the case of the localization of nucleosomes in isolated chromosomes in vitro. Template Learning outperformed the conventional template matching and classification routine typically employed for this type of task, both in terms of annotation Precision and orientational isotropy. Notably, all annotations generated by Template Learning could be directly used for achieving high-resolution STA without requiring any manual or classification-based curation.

While Template Learning has proven effective for annotating relatively large and structurally defined particles, such as ribosomes, FAS, and nucleosomes, smaller, less-defined, or highly flexible molecules present unique challenges. These challenges stem from inherent limitations in cryo-ET data quality and the increased complexity required to adequately represent such particles in simulations. Future advancements in cryo-ET hardware and software are likely to enable higher-resolution data acquisition. For flexible molecules, incorporating more diverse structural templates, potentially derived from molecular dynamics simulations or AlphaFold predictions, could significantly enhance Template Learning's applicability, albeit at the cost of additional computational overhead.

In future work, we envision adapting our framework to locate structures bound to other biomolecules or exhibiting specific protein-protein interactions, such as membrane and filamentous proteins, by allowing to incorporate additional priors on particle distributions into the domain randomization pipeline. Another challenge will be to reduce computational issues as physics-based simulations are still resource-intensive.

Conclusively, we believe Template Learning's versatile and straightforward framework marks it as a timely and potent tool for a broad spectrum of cryo-ET studies, ready to make significant impacts in the field.

## Methods

### Normal mode analysis (NMA)

Our study employed NMA on various atomic structures, prioritizing computational efficiency over precise molecular mechanics simulation, as the primary focus was on training the network on variations of the target biomolecule rather than precisely predicting its conformations. This emphasis on expanding the domain of the synthetic data aimed to capture a broad range of variations, not solely constrained to those witnessed in experimental data.

To manage computational demands, we adopted a coarse-grained modeling strategy[52], focusing on Carbon Alpha and Phosphorus atoms for NMA and subsequently expanding (interpolating) the modes to the complete atomic structure. Typically, a substantial number (three times the number of atoms in the atomic structure) of normal modes are calculated simultaneously, which can be ordered based on their frequency. Low-frequency normal modes are particularly useful as they represent global movements, while high-frequency normal modes depict local movements. Prior studies have consistently highlighted the significance of these low-frequency, high-collective modes in capturing experimentally observed conformational variabilities[51,53].

The user of NMA retains control over the selection of modes and the amplitude of deformation to generate flexible variations, ensuring a balance between simulating variability and maintaining structural integrity. As a guideline, we chose the first 20 low-frequency normal modes for generating random flexible variations. Setting the amplitude range to 150 for ribosomes and 100 for nucleosomes resulted in Root Mean Square Deviation (RMSD) values of approximately 1 Å and 2 Å from the initial structures, respectively. Adjusting this parameter incurs minimal computational costs (a few minutes) by generating variations and assessing the RMSD from the initial structure, for instance, using ChimeraX[54].

The computational time required for the calculation of NMA varies depending on the size of the input atomic structure. Specifically, for an atomic structure of 200 kDa, such as a nucleosome, the calculation typically takes approximately 1 min. In contrast, for larger structures like a ribosome weighing around 4.5 MDa, the calculation time extends to about 1 h when processed on a single CPU (benchmarked on Intel(R) Xeon(R) W-2145 CPU @ 3.70 GHz).

In this work, all NMA calculations were performed using ProDy[55], a widely recognized open-source Python package for molecular mechanics simulations.

### Tetris algorithm for generating highly crowded samples

The Tetris algorithm (flowchart shown in Fig. 2) utilizes input volumes of the biomolecules, essentially volumetric versions of the PDB templates and distractors, to generate a densely packed sample comprising randomly rotated biomolecule copies. It systematically places one randomly rotated molecule at a time in the closest proximity to previously positioned molecules. The algorithm terminates when a new copy cannot be placed in the sample.

In the initial step, the Tetris algorithm places a molecule at the center of a larger volumetric sample, the size of which is set by the user, establishing the current Tetris output. Subsequent iterations involve randomly rotating a new molecule, followed by its conversion to binary form after applying low-pass filtering and thresholding. While the values for low-pass filtering and the volume threshold are empirical and subject to variation, a standard deviation sigma of 2 for a Gaussian low-pass filter, and a volume threshold of 100 have generally proven effective in our trials (note that the volumes are generated from PDBs using Eman2[47] e2pdb2mrc software). This copy (molecule) is then diluted using a ball-like structural element of user-defined radius. Multiplying the output of the binarized volume by a large positive number (integer infinity in programming) and subtracting the result from the dilated volume, results in an "insertion shell", which serves as a spatial guide for placing the next molecule. Notably, the radius of the structural element provides direct control over the compactness of the sample.

The correlation between this insertion shell and a binary version of the current output (the binary version is smoothed and thresholded the same way as above) generates a correlation map. White voxels in the correlation map represent potential locations for adding the copy, ensuring the desired intermolecular distance. In contrast, black and gray voxels represent locations where the copy would intersect with previous copies or be too distant from other molecules, respectively. If the correlation map is not entirely negative (stopping criteria), the current copy is added at the index of the maximum entry, corresponding to the location closest to the sample within the desired distance.

During Tetris sample generation, the oriented coordinates used for placing molecules in the volumetric sample are preserved. To generate cryo-ET simulations of this sample, the oriented coordinates are passed to the physical-based simulator (Parakeet).

Generating volumetric samples using the Tetris algorithm with molecules at a resolution of 32 Å and a voxel size of 16 Å$^3$, as implemented in our current software, achieves a placement rate of approximately 1000 molecules per minute. The creation of a typical Tetris, with dimensions of $3072 \times 3072 \times 1024$ Å$^3$, requires approximately 5 to 6 min on a single CPU (benchmarked on Intel(R) Xeon(R) W-2145 CPU @ 3.70 GHz) and can accommodate around 5000 molecules. Consequently, the generation of 48 Tetris samples necessary for the tests performed in this study takes approximately 4.5 h on a single workstation.

### Performance assessment metrics of particle picking

In evaluating how well our deep learning models perform in particle localization, we employed standard metrics such as Precision, Recall, and $F_1$ score. Precision measures the percentage of correct picks out of all the particles selected by the model (True Positives / Positives), while Recall measures how many of the actual particles were correctly identified (True Positives / Ground Truth). The $F_1$ score, as the harmonic mean of Precision and Recall, offers a balanced measure of overall model performance.

DeepFinder, as a segmentation-based particle annotation method, assigns categorical probabilities to each voxel, predicting whether it belongs to the background or a target class. Each voxel is assigned two categorical probabilities based on the model's confidence (or more if multiple targets are assigned simultaneously, though this was not used in our work). These probabilities are combined to form the final segmentation map: a voxel is assigned to the background (0) if its background probability is higher, or to the target class (1) if its target probability is higher. This voxel-wise segmentation manifests as a complex landscape of segmentation events ranging from complete and partial segmentation of true positives to incorrect segmentation of other molecules and background regions, which makes threshold setting essential. The optimal threshold should minimize false positives while sacrificing only a small number of true positives, achieving a balance between precision and recall for a high overall $F_1$ score.

The recall and precision curves in DeepFinder (e.g., in Figs. 3 and 4) are derived from the size of these segments. False positives typically receive lower scores, appearing toward the left side of the curve, where the threshold eliminates low-confidence segments, causing an initial rise in precision. However, at very high thresholds, more true positives are excluded than large volume segmented false positives, which explains why precision drops at the far right of the curve.

Similarly, recall starts high because even low-confidence, partially segmented true positives are included before thresholding. As the threshold increases, not only are false positives removed, but also these partially segmented true positives, leading to a decrease in recall.

In our experiments, we determined segment size thresholds that balanced precision and recall based on ground truth (expert-validated) annotations. We optimized the threshold for each experiment individually to ensure a meaningful ablation study and close-to-optimal performance for each model, allowing a fair comparison with other methods such as DeePiCt, DeepFinder, and Template Matching. However, our results showed that in several Template Learning variations, especially those involving multiple PDB templates and flexibility, the $F_1$ score remained relatively stable near the "sweet spot"—where recall and precision curves intersect—across a range of thresholds (see Supplementary Note 2 - Supplementary Figs. 10 and 11).

### Converting volumes to pseudoatoms with contrast tuning

To convert a volume to a pseudoatomic structure that allows physics-based simulations, we devised a two-stage algorithm. In the initial stage, we binarize the volume, similar to the process of generating tight 3D masks in cryo-ET and cryo-EM studies. Subsequently, we assign a pseudoatom to the position of every non-zero voxel, resulting in a point cloud as output. We parse this point cloud into a PDB format, representing each point (i.e., pseudoatom) as a density (DENS line in PDB format). Then, the resulting file is ready to be integrated into the physics-based simulations. However, projections of this structure lack the desired contrast compared to those observed in experimental data. This discrepancy arises from the inherent modeling of electron-atom interactions in physics-based simulators. To address this contrast issue, in the second stage, we substitute a fraction of pseudoatoms with actual atoms, such as phosphorus (e.g., replacing one out of three pseudoatoms with phosphorus atoms). This substitution is not aimed at replicating the precise chemical composition of the structure (a direction that may be explored in future research) but rather to empirically enhance contrast. We determine this conversion ratio (i.e., pseudoatoms to phosphorus) by projecting the pseudoatomic model in 2D and comparing it to the experimental data targeted for analysis.

### Cryo-ET of partially decondensed mitotic chromosomes

Chicken DT40 cells were maintained at a density of 8-10 ×105 /ml at 39 °C and 5% $CO_2$. The cells were synchronized to mitosis using 0.5 μg/ml of nocodazole for 13 h, resulting in a mitotic index of 70–90%. The mitotic chromosome isolation was performed using the classic polyamine-EDTA buffer-based method optimized for DT40[56]. The synchronized cells were harvested by centrifuging the culture at 1600 × g for 5 mins at 4 °C. The cells were swollen at room temperature for 5 min in a low salt buffer in the presence of polyamines and then lysed using a Dounce homogenizer. The lysate was then overlaid on a step sucrose gradient (15%, 60%, and 80% [w/v]) centrifugation. The 60–80% interface was recovered and sedimented on a self-forming Percoll gradient in the presence of polyamine. The band containing the chromosomes was recovered and washed to remove the excess of Percoll. The isolated chromosomes are stored in Tris-HCl buffer (pH 7.5) containing polyamines in 60% glycerol.

The DT40 chromosomes were vitrified in glow-discharged (operating at 30% power with a gas mixture of 80% Argon:20% Oxygen) 200 copper mesh, Quantifoil Multi Holes grids (Quantifoil Micro Tools GmbH, Germany).

The isolated mitotic chromosomes were exposed to a low ionic strength buffer by 100-fold dilution in an aqueous solution containing TEEN buffer (1 mM Triethanolamine:HCl, pH 8.5, 0.2 mM Na-EDTA, pH 9 and 10 mM NaCl) for 1 h at 4 °C. This resulted in swelling of the mitotic chromosomes, which were then carefully spun down onto the surface of the grid. 10 nm BSA Gold Tracer (EMS, Hatfield, PA, USA) was added to the grid at a 5:1 ratio (chromosome:gold). The grid was blotted from the carbon side using a Teflon sheet and the metal side using blotting paper, respectively. The grid was blotted for 25 seconds with a blot force of 10 and flash-frozen into liquid ethane using Vitrobot Mark IV (FEI) at 4 °C and 100 % humidity.

The tilt series was recorded on Titan Krios G3 (Thermo Fisher Scientific) equipped with a Quantum energy filter with a slit width 20 eV for higher magnifications and a Gatan K2 detector using SerialEM Version 4.1.0 beta[57]. The images were acquired at a pixel size of 2.075 Å/pixel at 2–4.0 μm defocus at a dose rate of 3.2 e⁻/Å² per image fractionated over 10 frames. A dose symmetric tilt scheme[58] was used with a 3-degree increment step, and the tilt range was set to ±60 degree using SerialEM.

Individual movies are motion-corrected and averaged to form a tilt series using MotionCor2[59]. Gctf was used to perform CTF exstimation[60]. The tilt series were aligned and processed with IMOD[48]. The position of the grid in each image is aligned by tracing 10 nm gold particles as fiducials throughout the tilt series. The aligned tilt series were reconstructed into 3D tomograms with weighted back-projection with simultaneous iterative reconstruction technique (SIRT)-like filter after 4-fold binning.

After picking nucleosomes with our Template Learning trained DeepFinder model, we performed a reference-free STA following the procedure suggested in Relion V4.

### Reporting summary

Further information on research design is available in the Nature Portfolio Reporting Summary linked to this article.

## Data availability

The tilt series, tomogram, coordinates, and other metadata necessary to reproduce the nucleosome subtomogram averaging are deposited on EMPIAR via the accession code EMPIAR-11969). The subtomogram averages for nucleosomes in vitro are available on EMDB via the accession codes EMD-19823 and EMD-19825. The subtomogram averages for ribosomes in situ (extracted and averaged DEF tomograms of EMPIAR-10988) are available via the accession codes EMD-51648, EMD-51651, EMD-51652, and EMD-51653. The subtomogram averages for nucleosomes in situ are available via the accession codes EMD-51694, EMD-51695, and EMD-51696. Ribosome, FAS and nucleosome atomic structures used to simulate data are publicly available on PDB: 4UG0, 4V6X, 6Y0G, 6Y2L, 6Z6M, 7CPU), 4V59, 6QL5), 2PYO, 7KBE, 7PEX, 7PEY, 7XZY, and 7Y00. The PDB IDs for the distrctors used in Fig. 1 are 3QM1, 7NIU, and 6UP6. The material for reproducing the results within the Figures and Supplementary Figures is available in the Source Data file. Source data are provided in this paper.

## Code availability

Template Learning code[61] is freeware, open source, GPU and CPU optimized, and fully implemented in Python (https://github.com/MohamadHarastani/TemplateLearning). It provides all the steps that are required to reproduce the results presented in this article and can be straightforwardly applied to other studies. It provides wrappers for ProDy, Eman2 and Parakeet that will be installed during the installation. The software was tested on our workstations (Dell Precision 5820, Intel(R) Xeon(R) W-2145 CPU @ 3.70 GHz, 96 GB DDR4 RAM, 2 X NVIDIA RTX A6000 or NVIDIA Quadro RTX 8000). The time required

to perform the Template Learning workflow ranged from 1 to 2 days (depending on the number of used GPUs). Installation and user guides are available on GitHub.

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

## Acknowledgements

We thank Daesik Jang from Huawei Technologies Canada for discussions on the synthetic-to-real domain gap, Victor Hanss, Fadwa Fatmaoui, Nils Marechal, and Alexandre Durand from the Center for Integrative Biology (CBI) in Illkirch, France, for their technical support, discussions, and assistance with sample preparation and data collection, Emmanuel Moebel from Inria, Rennes, France, for help with DeepFinder, James M. Parkhurst from the Rosalind Franklin Institute, UK, for help with Parakeet, and Wim Hagen for data collection support at EMBL in Heidelberg, Germany. The work was funded by ANR (ANR-20-CE11-0020-02 to M.H. and M.E., ANR-23-CE45-0012-01 to M.E., ANR-23-CE45-0012-03 to C.K.), LabEx (ANR-10-LABEX-30-ME to G.P. and M.E.), the French Investments for the Future Program (ANR-17-EURE-0023), and iNEXT PID 22849, Horizon 2020, FRISBI, and Instruct-ERIC.

## Author contributions

Investigation, conceptualization, and methodology were done by M.H. and M.E. with contributions from all authors. Software programming and computational experiments were done by M.H. Sample preparation and data acquisition were done by G.P. under the supervision of ME. STA was done by M.H. and G.P. under the supervision of M.E. Validation and results interpretation were done by all authors. Writing the original draft was done by M.H. with input from G.P. Review and editing were done by M.H., M.E., and C.K. Funding acquisition was done by M.E. and C.K. Project administration was done by M.E.

## Competing interests

The authors declare no competing of interest.
