## [Transparent Peer Review file · Nature Communications]

Template Learning: Deep Learning with Domain Randomization for Particle Picking in Cryo-Electron Tomography

Corresponding Author: Dr Mohamad Harastani

Version 0:

Reviewer comments:

Reviewer #1

(Remarks to the Author)

This manuscript proposes Template Learning for particle picking in cryo-electron tomography (cryo-ET). Template Learning integrates deep learning's accuracy in particle identification with the ability to train models on biomolecular templates through domain randomization. This method automates the simulation of training datasets, addressing issues like molecular crowding, structural variability, and data acquisition variations. Overall there is a lack of methodological innovation.

I have several specific comments from template learning pipeline design and experiments:

Model:

1. The application of domain randomization in cryo-ET data analysis is not new and was previously introduced by Che et al. [48]. This paper fails to sufficiently differentiate its approach from Che et al.'s work or discuss in detail the novel contributions it makes beyond existing methodologies. Without clear distinctions and advancements over previous work, the technical novelty of this paper remains questionable.

2. The study could benefit from a deeper exploration of the interactions between target particles and distractors. While the current experiments include configurations with no distractors and fewer distractors, they do not sufficiently address scenarios where distractors are not just only physically similar to target particles but also share similar feature representations as interpreted by CNNs. These "hard samples" (distractors might be similar to target particles) are crucial for probing the model's limits, as such distractors would significantly increase both false positives (by leading to incorrect identification of distractors as targets) and false negatives (by missing real targets mistaken for distractors). Conversely, if the model successfully learns to differentiate between very similar targets and distractors, it might generalize better to new datasets, assuming these also contain closely similar particles. This should be further validated, with more experiments about target particles vs. distractors.

3. Additionally, the original DeepFinder work enhances model robustness by incorporating mislabeled distractors into training data. This built-in robustness helps the model learn to ignore these incorrect labels, enhancing its ability to focus on accurately identified features of macromolecules. Does this study similarly integrate label noise into its training process? Please clarify.

Experiments:

1. There is a discrepancy in the evaluation methodology. As mentioned in lines 266-280, baselines used 2 out of 10 tomograms for evaluation, while Template Learning used all 10 tomograms. This inconsistency should be explained. To ensure a rigorous comparison, the same test set should be used in both algorithms.

2. Fig. 6 underscores that while fine-tuning substantially improves model performance, however, it also highlights potential shortcomings in the original model's generalizability. The need for fine-tuning suggests that the initial training may not fully capture the variabilities inherent in different sample preparations, electron beam settings, detector types, and biological specimen characteristics. The effectiveness of fine-tuning is heavily dependent on the quality and representativeness of the

new data; poorly representative fine-tuning data can introduce new biases or lead to specific dataset overfitting. However, the well-annotated tomograms for fine-tuning the target task might be insufficient. To mitigate these issues, the authors should consider additional strategies to enhance the original model's generalizability.

3. The experimental results, especially Fig. 4, require additional explanation to fully understand the observed trends in precision and recall. Specifically: i) The author should clarify the underlying reasons why precision initially increases and then decreases, and why recall consistently drops across different setups. Please give more detailed explanations. ii) It is noteworthy that in the no distractor setting (Fig. 4-E), the recall does not decrease as significantly as in other experiments. This observation might suggest that the presence of distractors heavily influences the model's ability to accurately identify target particles. The authors should discuss whether this implies the model is overly sensitive to distractors, potentially mistaking them for targets in more complex settings. iii) The authors should incorporate statistical tests to evaluate the significance of the observed differences in performance with different experimental settings could lend more credibility to the conclusions drawn from these experiments. iv) The color scheme in Fig. 4 would cause misunderstanding, while blue stands for F1 (-mask) and precision (+mask) simultaneously, and red stands for recall and F1 (+mask) simultaneously.

(Remarks on code availability)

The author provided README and short tutorials. I didn't try to install and run the code.

Reviewer #2

(Remarks to the Author)

In this manuscript the authors present a clever method to overcome the "synthetic-to-real domain gap" for the generation of training data for deep learning methods to pick particles in Cryo-ET images. The presented method – called Template Learning - is an effective approach to generate such training data fully automatised, thus omitting the tedious step of manual annotation. Moreover, it is applicable across domains (again mitigating the need for manual annotation).

To account for the various sources of divergence between synthetic and real data, approaches to handle molecular crowding, structural variabilities, and data acquisition variations are included. To ensure the applicability in real world scenarios not only known structures from the PDB, but also volumetric templates (e.g. experimentally retrieved volumes from partially annotated lower resolution subtomograms) can be used to create the training data.

Skipping the need for manual annotations and the applicability across domains renders it a mighty tool, that potentially changes the Cryo-ET field, as currently in most experiments the annotation step is the most time consuming. Omitting this step, will speed up experiment evaluation and scientific discovery.

The dataset selected for the study, an exhaustively annotated in situ Cryo-ET dataset, is appropriate for testing a Deep Learning method for Cryo-ET data, especially since comparisons to other methods are available for that dataset. The experiments are designed nicely to validate the method, an ablation study was done to justify all components of the proposed method, the domain adaption was demonstrated on defocused tomography data (same target molecule, but different image acquisition), fine tuning was demonstrated by using appropriate experimental data (different target molecule), and an example application on a completely different dataset molecule was done. The conclusions drawn from these experiments are well supported by the presented experiments.

The quality of the presentation is good, experiments are explained in detail, results are given in adequate overview graphs. Overall, the text is accessible and concise, without skipping over important steps that are relevant for understanding. In the introduction a good context on the state of deep learning on Cryo-ET data is given, outlining nicely the new contributions of the method.

The experimental set-up is explained sufficiently and results are described and supplemented with clear overview graphs, that only need minor fine-tuning (see below). In summary the presented work is original and significant.

Suggested improvements:

1. There are some minor issues in Figure 3, regarding visibility. In Fig. 3 I advise to change the colours to something more readable (do not use these shades of green and yellow, as they are hard to differentiate), and also increase the brush stroke for circling TP/FP/FN and to indicate zoom-region.

2. Especially, Fig. 4 and Fig. 5 A/B need some refinement, as it is not yet easy to understand and might be misleading.

a) Add axis labels to x and y, and also indicate the unit of the x-axis.

b) In Fig. 4 A-H and Fig. 5A-B, plots are referred to as "recall-precision plots". Under this term usually plots of precision vs. recall are expected. I suggest to use another name.

c) Since for the F1 score the median from the different tomograms is reported, I am wondering if precision and recall are the median values as well (if so, please include error bars), or if these are computed over all tomograms jointly (if so please clarify this, and provide the overall F1 corresponding to these numbers).

d) The plots depicting the F1 spread are missing an indication at which exact segment volume size they are taken (maybe indicate in the recall and precision plot).

3. When investigating the different Template Learning variations, please discuss why the segment size threshold varies for different experiments (Fig. 4). Should this not relate to the volume of the searched protein complex (which is always

ribosome in these experiments)? How do I get the sweet spot of the “best F1” when I do not have a ground truth for segment volume determination?

4. Generally, a short discussion on why using (cytosol) masks is beneficial, is missing. A plot in the extended data section showing which features outside the mask are falsely identified, resulting in such a big impact on the F1 score, could be interesting for future users of the method.

5. In Extended Data Fig. 3 the used segment size cut-off is missing. In addition, I consider a colour coding of TP and FP in these plots beneficial.

6. I am wondering to which extend partially known structures and/or (in times of alpha fold) theoretical models are sufficient. However, this experiment should probably be done by the future users and is not necessary for this paper.

(Remarks on code availability)

I did not do an in depth code review. Based on a brief look on it, it looks reasonable, well documented, and probably usable.

Reviewer #3

(Remarks to the Author)

(Remarks on code availability)

Reviewer #4

(Remarks to the Author)

Deep learning has become a powerful tool for streamlining particle picking in cryo-electron tomography (cryo-ET) workflows. However, conventional algorithms often rely on supervised learning, requiring substantial manual annotation of training data – a significant bottleneck. This paper introduces Template Learning, a new technique that tackles this challenge. It generates simulated training datasets compatible with existing deep learning approaches. By employing domain randomization, Template Learning bridges the gap between synthetic and real-world data. It incorporates factors like molecular crowding, structural variations, and data acquisition variability to mimic real cryo-ET datasets.

The authors benchmark Template Learning, demonstrating its potential to significantly reduce or even eliminate the need for manually annotated training data. Notably, they successfully trained a deep learning model for ribosomes using simulated datasets generated by Template Learning. This model outperformed models trained on manually labeled experimental data for ribosomes.

This work presents a promising approach with the potential to significantly impact CryoET workflows. Addressing the points raised below will further strengthen the work and demonstrate the broad applicability of Template Learning.

Major points:

The ability to train a deep learning model for ribosomes using only simulated datasets generated by Template Learning is an impressive achievement. This model's superior performance compared to those trained on experimental data for ribosomes is highly promising. To further validate the approach, subtomogram averaging of the picked ribosomes should be compared to results obtained from models trained on experimental tomograms. This would provide a more comprehensive assessment of the quality of the picked particles.

The approach's generalizability to non-ribosomal targets, particularly within crowded cellular environments and without volta phase plates, requires further demonstration. For example, the authors mention a median F1 score of 0 for FAS annotation on DEF datasets when using only simulated data. Can the authors detail how much fine-tuning with experimental data is necessary when simulated data alone performs poorly? This would provide valuable insights into the practical application of Template Learning for various targets, particularly in crowded cellular environments and without volta phase plates.

To strengthen the case for broader applicability, the authors could present an additional non-ribosomal example from a cellular environment. Ideally, this example would demonstrate models trained on simulated data either outperforming or performing equally well as those trained on experimental data. This would provide a stronger demonstration of the approach's potential for a broader application.

Template Learning surpasses traditional 3D template matching in picking precision and offers better orientational isotropy for annotating nucleosomes from isolated chromosomes. Applying the trained nucleosome model to publicly available datasets like EMPIAR-10678 or EMPIAR-10179 would strengthen the case for generalizability of the trained model.

Can Template Learning be effectively used for filamentous macromolecules? If so, testing the model's annotation capability on a dataset like EMPIAR-10989 would be a valuable addition.

Minor points

For clarity, the text should differentiate between 2D and 3D template matching.

The paper doesn't explicitly address how compositional heterogeneity, more specifically protein-protein interactions, is handled by Template Learning. Clarifying how the method accounts for these variations would be beneficial.

The generation process of the "cytosol masks" used in the study should be described. Understanding how these masks are derived is important for interpreting the results.

(Remarks on code availability)

Version 1:

Reviewer comments:

Reviewer #1

(Remarks to the Author)

Thanks to the authors' responses, these responses partially solve my questions. However, there are still some follow-up concerns:

1. While Template Learning falls within the template-matching domain for particle picking, its dependency on high-quality templates could pose limitations, particularly in cases where atomic structures are unavailable—a common situation in in situ cryo-ET data analysis. Could the authors provide more insights into how lower-resolution templates (e.g., from low-resolution EM maps) impact model performance and what adjustments, if any, might be necessary for such cases?

2. The method was tested on ribosomes, nucleosomes, and FAS, which are relatively large and structurally defined particles. However, smaller, less-defined particles or highly flexible molecules might perform less robustly with Template Learning. It would be helpful if the authors could discuss the method's limitations for these particle types and clarify if specific adjustments are required for handling diverse macromolecular structures.

3. Template Learning is proposed as a solution to the orientation biases typically seen in template matching. Could the authors elaborate on how this method specifically mitigates orientation biases, especially when applied to asymmetric particles?

4. I appreciate the authors' acknowledgment of the influence of expert-validated annotations on performance curves, which underlines the need for subtomogram averaging to validate results. However, as shown in Figure 7, the improvement from traditional template matching to Template Learning in subtomogram averaging resolution appears marginal (12.9 Å to 12.8 Å).

(Remarks on code availability)

The author provided README and short tutorials. I didn't try to install and run the code.

Reviewer #2

(Remarks to the Author)

Dear Authors,

all my concerns have been addressed appropriately. I am happy to recommend "Template Learning: Deep Learning with Domain Randomization for Particle Picking in Cryo-Electron Tomography" for publication in Nature Communications.

(Remarks on code availability)

Reviewer #3

(Remarks to the Author)

(Remarks on code availability)

Reviewer #4

(Remarks to the Author)

The revisions have addressed my previous concerns and significantly improved the manuscript. I am happy to recommend it for publication.

(Remarks on code availability)

Version 3:

Reviewer comments:

Reviewer #1

(Remarks to the Author)

All my concerns are addressed. I want to thank the authors for their efforts.

(Remarks on code availability)

Reviewer #2

(Remarks to the Author)

I still recommend this manuscript for publication. Good Luck.

(Remarks on code availability)

Reviewer #3

(Remarks to the Author)

(Remarks on code availability)

Reviewer #4

(Remarks to the Author)

I appreciate the addition of the volume to pseudoatomic model conversion algorithm. However, I don't currently see it on the GitHub page. Could you add it?

I am happy to recommend the paper for publication.

(Remarks on code availability)

RESPONSES TO REVIEWERS' COMMENTS

We thank the reviewers for their insightful feedback, which has been fully addressed in our revised manuscript. Below is a concise summary of the main changes:

- We addressed all reviewer feedback regarding citation of relevant literature, statistical tests on the benchmarking results, figure quality, and expanded discussions (**Supplementary Notes 1-2**, main text **Figures 3-6**, **Extended Data Figure 3**).
- Acknowledging that introducing distractors is a new domain randomization axis for cryo-ET applications, we validated the robustness of our pipeline by demonstrating that it maintains state-of-the-art performance even after mislabeling more than 10% of the simulated data during model training (**Supplementary Note 3**).
- Supporting the findings of Template Learning's state-of-the-art performance with ribosome subtomogram averaging, we showed that Template Learning annotated 6% more ribosomes, and the quality of ribosome particles picked by Template Learning was superior for subtomogram averaging compared to those annotated by experts but missed by Template Learning (**Supplementary Note 4**).
- We added a new set of experiments comparing nucleosome annotation *in situ* using Template Learning, supervised deep learning, and template matching. The results demonstrate superior precision and angular distribution of annotations from Template Learning (**Supplementary Note 5**).

All **changes in the main manuscript**, aside from the points above, are highlighted in green.

Reviewers' Comments:

Reviewer #1 (Remarks to the Author):

This manuscript proposes Template Learning for particle picking in cryo-electron tomography (cryo-ET). Template Learning integrates deep learning's accuracy in particle identification with the ability to train models on biomolecular templates through domain randomization. This method automates the simulation of training datasets, addressing issues like molecular crowding, structural variability, and data acquisition variations. Overall there is a lack of methodological innovation.

We would like to thank Reviewer #1 for their constructive feedback and for helping us improve the quality of our manuscript. While we appreciate the reviewer's insights, we respectfully disagree with their assessment regarding the lack of methodological innovation. Below, we provide a point-per-point response to the reviewer's comments and a detailed clarification of the key innovations introduced in our approach.

I have several specific comments from template learning pipeline design and experiments:

Model:

1. The application of domain randomization in cryo-ET data analysis is not new and was previously introduced by Che et al. [48]. This paper fails to sufficiently differentiate its approach from Che et al.'s work or discuss in detail the novel contributions it makes beyond existing methodologies. Without clear distinctions and advancements over previous work, the technical novelty of this paper remains questionable.

We appreciate the reviewer highlighting the relevance of our work in relation to the research by Che et al., initially presented in conference proceedings. Importantly, the same research group, the Xu lab, extended this work in a journal article titled Cryo-Shift (Bandyopadhyay et al., 2021). In our manuscript, we have already dedicated a paragraph to Cryo-Shift in the Introduction and cited Che et al. in our discussion. In the revised manuscript, we cited Che et al. in both Introduction and Results and added a discussion on their methodology in Conclusion.

Our approach, Template Learning, introduces several key differences compared to the methods described by Che et al. and Bandyopadhyay et al. In their work, particle picking was performed prior to analysis, and macromolecules were extracted as isolated subtomograms (Che et al., section IV.B). Specifically, these extracted subtomograms contained four predefined types of macromolecules/structures: 80 ribosomes, 460 mitochondrial membranes, 125 TRiCs, and 386 single-capped proteasomes.

In contrast, our study demonstrates that the Template Learning pipeline is a robust, ready-to-use tool for cryo-ET studies, capable of identifying target molecules in crowded native cellular environments containing hundreds or even thousands of different macromolecules. To address the increased task complexity, we incorporated several key innovations into our methodology.

Che et al., and subsequently Bandyopadhyay et al., applied domain randomization by simulating acquisition parameters of isolated macromolecules. Their simulations randomized variables such as signal-to-noise ratio, missing wedge range, defocii, and spherical aberrations. While these domain randomization axes can be adequate for processing predefined and isolated subtomograms, **our research introduces two novel domain randomization concepts essential for molecular identification in experimental tomograms: distractors and crowding**. These elements are core innovations in our methodology and software, representing new domain randomization axes that were not previously explored for particle picking in cellular cryo-ET data.

We found that including distractors—non-specific macromolecules in training sets—improved the F1 score by 22% for masked volumes (excluding noisy regions) and by 61% for raw reconstructions. This improvement is especially significant in raw reconstructions, where the presence of distractors greatly reduces the false positive rate in particle picking. Additionally, simulating high levels of crowding enhanced the F1 score by 15% in masked volumes and by 25% in complete reconstructions, a key factor without which the method would not have achieved its state-of-the-art performance. These detailed results are presented in Fig. 4 (A, E, and G).

Furthermore, we introduced improvements on several methodological axes that distinguish our work from prior studies:

1- Advanced and Efficient Physics-Based Simulations: We utilized a specialized cryo-ET simulator (Parakeet), capable of incorporating additional parameters, that are critical in bridging the synthetic-to-real domain gap such as dose-dependent damage (i.e., simulating dose symmetric acquisition scheme) and Volta phase plates (VPP), which were not considered in earlier works. Additionally, our software enables efficient simulation of complete tomogram sets in parallel using GPU acceleration, and outputs data in a format compatible with deep learning methods (e.g., DeepFinder), including tomograms with corresponding ground truth particle segmentation and annotation.

2- Template Diversity and Flexibility: While Che et al. and Bandyopadhyay et al. used a single template and addressed molecular flexibility through data augmentation (i.e., warping), our approach tackles compositional variability by allowing the incorporation of multiple target template structures—something that cannot be achieved with warping alone. Our approach also addresses molecular flexibility more realistically using coarse-grained Normal Mode Analysis, which generates structural variations at atomic scale. Our ablation experiments demonstrated that restricting the template to a single structure reduced the F1 score by approximately 10%, while incorporating multiple templates and flexible variations each improved performance by 5%, with a combined improvement of 10%.

3- Simulation of Crowding: We have introduced an algorithm we termed the “Tetris Algorithm”, that can efficiently simulate highly crowded multi-molecular environments without resorting to intricate molecular dynamics simulations. This enables us to capture cellular crowding without excessive computational overhead.

These innovations collectively contribute to the state-of-the-art performance of Template Learning, as validated on multiple independent benchmarks, including both *in situ* and *in vitro* cryo-ET data. Each of these contributions was rigorously tested through ablation experiments, confirming their importance in enhancing the performance and applicability of our method.

2. The study could benefit from a deeper exploration of the interactions between target particles and distractors. While the current experiments include configurations with no distractors and fewer distractors, they do not sufficiently address scenarios where distractors are not just only physically similar to target particles but also share similar feature representations as interpreted by CNNs. These "hard samples" (distractors might be similar to target particles) are crucial for probing the model's limits, as such distractors would significantly increase both false positives (by leading to incorrect identification of distractors as targets) and false negatives (by missing real targets mistaken for distractors). Conversely, if the model successfully learns to differentiate between very similar targets and distractors, it might generalize better to new datasets, assuming these also contain closely similar particles. This should be further validated, with more experiments about target particles vs. distractors.

We thank the reviewer for raising this important point about the potential influence of target-distractor similarity on Template Learning.

In our method, we selected a set of 100 structures with general shapes and sizes as distractors, based on structures used in previous works such as the SHREC tomography competition (Gubins et al.) and TomoTwin (Rice et al.) (see Extended Data Fig. 1 for the shapes and sizes). These structures were shown to have sufficiently different feature representations, as demonstrated by CNN-based models (e.g., TomoTwin) clustering their feature embeddings into separate classes. As these distractors are inherently diverse, a single target structure cannot be similar to all 100 distractors.

In our data simulation pipeline, each distractor occupies approximately 1% of the total volume density allocated to distractors in the simulated tomograms. The density of target molecules and distractors is balanced at a 50-50% ratio. For instance, during the simulation of training data for ribosome and FAS picking, one target molecule (ribosome or FAS) is placed for every five distractors (see Extended Data Fig. 2 for a visual example). If similar or identical structures are used as both targets and distractors, only about 0.5% label noise is introduced for each identical distractor. In a supplementary experiment, we show that even mislabeling more than 10% of the target structures as distractors does not significantly affect model performance.

Future work focused on annotating targets in specific conformations, even when they resemble other structures, could be valuable and may facilitate alignment and classification routines like those in Relion. However, this is beyond the scope of this manuscript, which aims to introduce Template Learning and the advantages of using a general array of distractors within a computationally efficient software framework.

Nevertheless, based on the reviewer's suggestion, we performed a pilot experiment to assess the influence of target-distractor similarity on Template Learning performance. In this experiment, all copies of 18 out of 156 ribosome structures were intentionally mislabeled as distractors (~750 out

of ~6500 ribosome copies were mislabeled as distractors). The goal of this is hypothetical scenario is to test the robustness of the method. As detailed in Supplementary Note 3, the Template Learning-trained DeepFinder model achieved a high F1 score (~82%), with only a minor drop (~3-4%) compared to the original method. Our statistical tests showed that this drop was either weak or non-statistically significant, and the performance remained in line with other supervised learning techniques (i.e., in the F1 score range of 79-83%).

We have included the full details of this experiment in the supplementary information for further reference.

3. Additionally, the original DeepFinder work enhances model robustness by incorporating mislabeled distractors into training data. This built-in robustness helps the model learn to ignore these incorrect labels, enhancing its ability to focus on accurately identified features of macromolecules. Does this study similarly integrate label noise into its training process? Please clarify.

In constructing training datasets for training DeepFinder on experimental tomograms, label noise can arise from using incorrectly labeled annotations or segmentations (Moebel et al., Nat Methods 2021). However, in the context of Template Learning, training datasets generated from simulations are free from label noise, as the ground truth positions and segmentations of target structures are accurately known.

In the original DeepFinder paper, the authors demonstrated that some label noise—primarily due to using particle sphere segmentations (where parts of the background close to target molecules are included because of the spherical segments)—did not significantly affect performance, especially when the target molecule was not too small. However, DeepFinder does not intentionally mislabel classes to improve robustness, as the reviewer inferred.

Our Template Learning software supports both shape and sphere segmentations but does not alter DeepFinder's default data handling or training strategies. As such, it does not intentionally introduce label noise through intentional mislabeling during training. As shown in a supplementary experiment (Supplementary Note 3), intentional mislabeling did not significantly impact the performance of our method. While it improved Precision at higher thresholds, the substantial decrease in Recall outweighed any perceived benefits.

Experiments:

1. There is a discrepancy in the evaluation methodology. As mentioned in lines 266-280, baselines used 2 out of 10 tomograms for evaluation, while Template Learning used all 10 tomograms. This inconsistency should be explained. To ensure a rigorous comparison, the same test set should be used in both algorithms.

We thank the reviewer for pointing out to this issue. Previous methods, such as DeePiCt (Nat Methods 2023), used a supervised learning approach that required training on a subset of the experimental dataset. Contrary to what may have been inferred from our figures, they did not evaluate performance using only two tomograms. Instead, they reported the median F1 score from three cross-validation experiments, where in each experiment, the data was randomly split

into 8 tomograms for training and 2 tomograms for evaluation. This 3-fold cross-validation serves as an approximation of exhaustive cross-validation (Kohavi, 1995).

In contrast, Template Learning does not require training on experimental data and thus does not involve splitting the dataset into training and validation sets. This allows us to evaluate the F1 score using all 10 tomograms at once, which is equivalent to an exhaustive cross-validation. This evaluation of Template Learning using all 10 tomograms aligns with how template matching performance was reported for the same dataset, where all tomograms were included in the evaluation (DeePiCt, Nat Methods 2023, Figure 3.e).

For experiments that involved fine-tuning using experimental data (Results - Section 3), we followed the same 3-fold cross-validation strategy used in previous methods to ensure consistency and a fair comparison.

In our revised manuscript, we improved Figures 4 and 6 to clarify where the median was reported for 3 cross-validation splits.

Reference:

Kohavi, R. (1995). A study of cross-validation and bootstrap for accuracy estimation and model selection. In Proceedings of the 14th International Joint Conference on Artificial Intelligence (IJCAI) (Vol. 2, pp. 1137-1143).

2. Fig. 6 underscores that while fine-tuning substantially improves model performance, however, it also highlights potential shortcomings in the original model's generalizability. The need for fine-tuning suggests that the initial training may not fully capture the variabilities inherent in different sample preparations, electron beam settings, detector types, and biological specimen characteristics. The effectiveness of fine-tuning is heavily dependent on the quality and representativeness of the new data; poorly representative fine-tuning data can introduce new biases or lead to specific dataset overfitting. However, the well-annotated tomograms for fine-tuning the target task might be insufficient. To mitigate these issues, the authors should consider additional strategies to enhance the original model's generalizability.

We thank the reviewer for their insightful comments regarding model generalizability and the role of fine-tuning.

In all of our Template Learning experiments for ribosomes, FAS, and nucleosomes, we used identical simulation conditions. Benchmarking for FAS and ribosome annotations was performed on the same volumes (EMPIAR 10988). Given the consistent simulation parameters and state-of-the-art performance achieved on ribosome annotations, our current results do not suggest any issues in capturing the sample preparation, electron beam settings, detector types, or biological specimen characteristics of this dataset.

FAS picking, as previously reported by DeePiCt (Nat Methods 2023), poses particular challenges due to its shell-like structure and low signal-to-noise ratio. However, as shown in Fig. 6, Template Learning for FAS picking, outperformed training DeepFinder on experimental tomograms, even before fine-tuning. Specifically, training DeepFinder on experimental tomograms for FAS picking

(EMPIAR 10988, VPP) resulted in a median F1 score of 11%, whereas training it on Template Learning simulations resulted in a median F1 score of 18%. Template Learning followed by fine-tuning outperformed DeePiCt and achieved state-of-the-art performance. Additionally, Template Learning showed a threefold improvement in the median F1 score for FAS picking compared to template matching.

We recognize, however, that while fine-tuning is effective, it can introduce the risk of overfitting or bias if the data is not representative, which is a broader challenge in supervised learning and not specific to our approach. We have incorporated this discussion into the Conclusion section of our revised manuscript.

3. The experimental results, especially Fig. 4, require additional explanation to fully understand the observed trends in precision and recall. Specifically: i) The author should clarify the underlying reasons why precision initially increases and then decreases, and why recall consistently drops across different setups. Please give more detailed explanations.

DeepFinder is a segmentation-based particle annotation method, designed to segment potential targets in tomograms and extract the centroids of different segments as potential particle positions. The score used to plot the Recall and Precision curves in DeepFinder is based on the size of these segments. False positives generally receive lower scores, which appear toward the left end of the curve. These curves are plotted by progressively eliminating potential positions below a certain threshold. As the threshold increases, many false positives are removed, which explains the initial rise in Precision. However, while false positives at the far right of the curve are less common, they still exist. As a result, at extremely high thresholds, Precision drops because some true positives get excluded while false positives corresponding to larger segments remain, reducing the true positive rate.

Similarly, Recall (sensitivity) initially starts high because, before thresholding small segments, true positives that are segmented with low confidence (i.e., partially segmented) are still counted in the data. But as the threshold gets higher, not only are false positives get removed, but also true positives of these partially segmented targets, leading to a decrease in Recall.

We provide an illustration below, and we have included this detailed explanation in the Methods of our revised manuscript.

We would also like to point out that while these curves are indicative of the method's performance, they are influenced by the accuracy of the original expert-validated annotations that were used as ground truth. We ensured that these evaluations were consistent with the methodologies used in prior evaluations of DeePiCt, DeepFinder, and Template Learning. However, upon further validations of the picked ribosomes using subtomogram averaging on EMPIAR-10988 (DEF), we found that some particles initially labeled as false positives (~6% of the original dataset) were actually true positives missed by the expert validators. Further details on the ribosome subtomogram averaging results are provided in Supplementary Note 4.

Caption: an illustration explaining DeepFinder Precision and Recall curve trends in relation to the thresholding based on the segment size. True positives are shown in green and False Positives in red.

ii) It is noteworthy that in the no distractor setting (Fig. 4-E), the recall does not decrease as significantly as in other experiments. This observation might suggest that the presence of distractors heavily influences the model's ability to accurately identify target particles. The authors should discuss whether this implies the model is overly sensitive to distractors, potentially mistaking them for targets in more complex settings.

Thank you for your observation. The reason Recall does not decrease drastically for the model trained without distractors is that its low selectivity leads to identifying nearly every density with high confidence, behaving similarly to unspecific size-based density thresholding (see the image below, part of Extended Data Fig. 3 in the revised manuscript). Based on this discussion, we added the following sentence to our revised manuscript: *"The absence of distractors during Template Learning results in models that lack specificity in identifying targets, often mistaking other objects for targets in experimental tomograms."*

Caption: Central slice of a VPP tomogram from EMPIAR-10988 with its expert-validated (ground truth) ribosome segmentation and the output segmentation maps using two Template Learning variations, with distractors and without distractors. The segmentation maps are displayed in their raw form (left) and color-coded (right) after extracting annotations from the centroids of the segments using MeanShift with a clustering radius of 10 voxels and applying a segment threshold based on the model's optimal performance for each experiment (balancing Precision and Recall). In the color-coded segmentation maps, green represents true positives, red represents false positives, blue represents false negatives, and white regions represent signals that did not influence the annotation, typically because they were smaller than the optimal threshold.

iii) The authors should incorporate statistical tests to evaluate the significance of the observed differences in performance with different experimental settings could lend more credibility to the conclusions drawn from these experiments.

We thank the reviewer for this valuable suggestion. In response, we have made the following improvements in the revised manuscript:

- Performed statistical tests to assess the significance of the observed trends in performance across experimental settings and discussed the Results in the manuscript.

- Enhanced all boxplots to display individual data points in addition to the quartiles, providing a clearer representation of the data distribution.
- Included descriptive statistics for all experiments, allowing evaluation based on not only the median (which was necessary for comparison with previously published results of other methods in the literature) but also the mean and min-max ranges.

The updated box plots are included in the main text, and the descriptive statistics and statistical tests are provided in Supplementary Note 2.

iv) The color scheme in Fig. 4 would cause misunderstanding, while blue stands for F1(-mask) and precision (+mask) simultaneously, and red stands for recall and F1(+mask) simultaneously.

We addressed this point in our revised manuscript.

Reviewer #1 (Remarks on code availability):

The author provided README and short tutorials. I didn't try to install and run the code.

We thank Reviewer #1 again for helping us to improve our manuscript through their questions and comments and we hope that our revised version meets their expectations.

Reviewer #2 (Remarks to the Author):

In this manuscript the authors present a clever method to overcome the "synthetic-to-real domain gap" for the generation of training data for deep learning methods to pick particles in Cryo-ET images. The presented method – called Template Learning - is an effective approach to generate such training data fully automatised, thus omitting the tedious step of manual annotation. Moreover, it is applicable across domains (again mitigating the need for manual annotation).

To account for the various sources of divergence between synthetic and real data, approaches to handle molecular crowding, structural variabilities, and data acquisition variations are included. To ensure the applicability in real world scenarios not only known structures from the PDB, but also volumetric templates (e.g. experimentally retrieved volumes from partially annotated lower resolution subtomograms) can be used to create the training data.

Skipping the need for manual annotations and the applicability across domains renders it a mighty tool, that potentially changes the Cryo-ET field, as currently in most experiments the annotation step is the most time consuming. Omitting this step, will speed up experiment evaluation and scientific discovery.

The dataset selected for the study, an exhaustively annotated in situ Cryo-ET dataset, is appropriate for testing a Deep Learning method for Cryo-ET data, especially since comparisons to other methods are available for that dataset. The experiments are designed nicely to validate the method, an ablation study was done to justify all components of the proposed method, the domain adaption was demonstrated on defocused tomography data (same target molecule, but different image acquisition), fine tuning was demonstrated by using appropriate experimental data (different target molecule), and an example application on a completely different dataset molecule was done. The conclusions drawn from these experiments are well supported by the presented experiments.

The quality of the presentation is good, experiments are explained in detail, results are given in adequate overview graphs. Overall, the text is accessible and concise, without skipping over important steps that are relevant for understanding. In the introduction a good context on the state of deep learning on Cryo-ET data is given, outlining nicely the new contributions of the method.

The experimental set-up is explained sufficiently and results are described and supplemented with clear overview graphs, that only need minor fine-tuning (see below). In summary the presented work is original and significant.

We would like to thank Reviewer #2 for reviewing our manuscript and for helping us improve the clarity of results and figures. We took into account all the suggested improvements and provided a point-per-point response below.

Suggested improvements:

1. There are some minor issues in Figure 3, regarding visibility. In Fig. 3 I advise to change the colours to something more readable (do not use these shades of green and yellow, as they are

hard to differentiate), and also increase the brush stroke for circling TP/FP/FN and to indicate zoom-region.

In our revised version, we used more readable colors and increased the brush strokes whenever applicable.

2. Especially, Fig. 4 and Fig. 5 A/B need some refinement, as it is not yet easy to understand and might be misleading.

a) Add axis labels to x and y, and also indicate the unit of the x-axis.

b) In Fig. 4 A-H and Fig. 5A-B, plots are referred to as “recall-precision plots”. Under this term usually plots of precision vs. recall are expected. I suggest to use another name.

c) Since for the F1 score the median from the different tomograms is reported, I am wondering if precision and recall are the median values as well (if so, please include error bars), or if these are computed over all tomograms jointly (if so please clarify this, and provide the overall F1 corresponding to these numbers).

d) The plots depicting the F1 spread are missing an indication at which exact segment volume size they are taken (maybe indicate in the recall and precision plot).

We thank the reviewer for pointing out these improvements, below are the improvements we made with respect to each point:

a) We reorganized these figures and added the axes and units as suggested.

b) We removed the usage of “Recall-Precision curves” and replaced “curves for Recall and Precision” to avoid the confusion of Recall-vs-Precision that it might indicate.

c) The Precision and Recall curves were computed over all the tomograms jointly. We added this clarification to our revised manuscript. We have also added the plots for the overall F1 score to Supplementary Note 2.

d) This threshold was always chosen where the Precision and Recall curves intersect, which often coincides with the highest overall F1 score (except for the curve shown in Fig 4.E (-mask), where the threshold was chosen based on the intersection point from the (+mask) case). We added this explanation in the corresponding captions of our revised manuscript. More information about the logic behind using this value is in the reply for the next point.

3. When investigating the different Template Learning variations, please discuss why the segment size threshold varies for different experiments (Fig. 4). Should this not relate to the volume of the searched protein complex (which is always ribosome in these experiments)? How do I get the sweet spot of the “best F1” when I do not have a ground truth for segment volume determination?

We greatly appreciate this insightful question. DeepFinder segments target particles by assigning categorical probabilities to each voxel, predicting whether it belongs to the background or a target class. Each voxel is assigned two categorical probabilities based on the model’s confidence (or

more, if multiple targets are to be assigned simultaneously; though possible, this was not used in our work). These probabilities are combined to form the final segmentation map: a voxel is assigned to the background (0) if its probability of being background is higher, or to the target class (1) if its target probability is higher. This voxel-wise segmentation manifests as a complex landscape of segmentation events ranging from complete and partial segmentation of true positives to incorrect segmentation of other molecules and background regions, which makes threshold setting essential. The ideal threshold minimizes false positives while sacrificing only a small portion of true positives, achieving a balance between Precision and Recall for a high overall F1 score.

Although the segment size relates to the volume of the searched protein complex, as inferred by the reviewer, a single threshold may not provide a fair comparison across different models. This is because their confidence levels, reflected in the segment sizes, vary depending on the nature of the training set they were exposed to. In our work, models were trained with different variations in their datasets to handle flexible molecules and diverse backgrounds, which caused voxel-level confidence—and therefore the optimal segment size threshold—to vary between models.

In our experiments, we determined segment size thresholds that give equal Precision and Recall (equal number of annotations between ground truth and model), based on the ground truth (expert-validated) annotations (see Illustration below).

Caption: Curves depicting the Precision, Recall, and F1 score of the typical Template Learning pipeline applied to annotate Ribosomes (in VPP tomograms of EMPIAR-10988).

We selected a threshold for each experiment individually to ensure a meaningful ablation study and close-to-optimal performance for each model, which also allowed a fair comparison with other methods such as DeepPiCt, DeepFinder, and Template Matching.

Moreover, our results show that for several Template Learning variations, especially those involving multiple PDB templates and flexibility, the F1 score remained relatively stable near the "sweet spot"—the point where Recall and Precision curves intersect—across a range of thresholds (see illustration above, also Supplementary Note 2 for more details).

As a rule of thumb, in the absence of ground truth annotations, using a threshold of 50% of the maximum segment size is usually valid. For example, in our ribosome picking benchmark experiments, the ribosome radius is roughly 10 pixels at a pixel size of 13.48 Å, and the maximum segment size, coating the ribosome, is around 11 pixels, corresponding to a spherical volume of approximately 5575 voxels³. Therefore, an optimal threshold should be around half of this value. A lower threshold favors higher recall, while a higher threshold favors higher precision.

We included a brief explanation in the Methods section of our revised manuscript detailing how DeepFinder operates and how to select an appropriate threshold based on this discussion.

4. Generally, a short discussion on why using (cytosol) masks is beneficial, is missing. A plot in the extended data section showing which features outside the mask are falsely identified, resulting in such a big impact on the F1 score, could be interesting for future users of the method.

Thank you for this valuable suggestion. In response, we have incorporated your feedback along with a suggestion from Reviewer #4. We added a discussion in Supplementary Note 1 that highlights the benefits of using cytosol masking, accompanied by Supplementary Figure 1.1, which illustrates examples of features falsely identified outside the mask. This discussion clarifies why removing these falsely identified particles leads to a significant improvement in the F1 score. Additionally, we have included information on how these masks can be generated both automatically and manually.

However, our results show that even in the absence of such masks, the performance of Template Learning remains remarkably high compared to the supervised methods evaluated (F1 score of Template Learning on VPP ribosome picking without/with mask: 0.79/0.85, compared to DeePiCt with mask: 0.79, and DeepFinder with mask: 0.83). Additionally, some of the "false positives" removed by masking could actually represent molecules that share structural similarities with the target particles, such as pre-ribosomal particles in the nucleolus or those captured in the nucleoplasm during their transport to the cytoplasm. Therefore, some of these particles, which are excluded by cytosol masking (see Supplementary Figure 1.1), may represent biologically relevant entities rather than non-specific false positives.

5. In Extended Data Fig. 3 the used segment size cut-off is missing. In addition, I consider a colour coding of TP and FP in these plots beneficial.

Thank you for these valuable comments, we updated the figure and its caption accordingly.

6. I am wondering to which extend partially known structures and/or (in times of alpha fold) theoretical models are sufficient. However, this experiment should probably be done by the future users and is not necessary for this paper.

Our current manuscript introduces the tool as a proof of concept, showing how atomic templates can be used to train deep learning models for cryo-ET particle picking through domain randomization-based simulations.

Our software supports the use of theoretical models, including those generated by AlphaFold, for data simulation and model training. As noted in our conclusion: "Template Learning can utilize any atomic models, including those generated by artificial intelligence-based tools, as templates."

We agree with the reviewer on the potential of our tool to assist in searching for theoretical predictions of proteins and protein complexes, and we look forward to supporting future users in achieving this goal.

Reviewer #2 (Remarks on code availability):

I did not do an in depth code review. Based on a brief look on it, it looks reasonable, well documented, and probably usable.

We thank Reviewer #2 again for allowing us to improve our manuscript through their comments and we hope that our revised version meets their expectations.

Reviewer #3 (Remarks to the Author):

We would like to thank Reviewer #3 for reviewing our manuscript and we hope that our revised version meets their expectations.

Reviewer #4 (Remarks to the Author):

Deep learning has become a powerful tool for streamlining particle picking in cryo-electron tomography (cryo-ET) workflows. However, conventional algorithms often rely on supervised learning, requiring substantial manual annotation of training data – a significant bottleneck. This paper introduces Template Learning, a new technique that tackles this challenge. It generates simulated training datasets compatible with existing deep learning approaches. By employing domain randomization, Template Learning bridges the gap between synthetic and real-world data. It incorporates factors like molecular crowding, structural variations, and data acquisition variability to mimic real cryo-ET datasets.

The authors benchmark Template Learning, demonstrating its potential to significantly reduce or even eliminate the need for manually annotated training data. Notably, they successfully trained a deep learning model for ribosomes using simulated datasets generated by Template Learning. This model outperformed models trained on manually labeled experimental data for ribosomes.

This work presents a promising approach with the potential to significantly impact CryoET workflows. Addressing the points raised below will further strengthen the work and demonstrate the broad applicability of Template Learning.

We would like to thank Reviewer #4 for reviewing our manuscript and for their valuable discussions and suggestions. We took into account all their comments and provided a point-per-point reply below.

Major points:

The ability to train a deep learning model for ribosomes using only simulated datasets generated by Template Learning is an impressive achievement. This model's superior performance compared to those trained on experimental data for ribosomes is highly promising. To further validate the approach, subtomogram averaging of the picked ribosomes should be compared to results obtained from models trained on experimental tomograms. This would provide a more comprehensive assessment of the quality of the picked particles.

Thank you for highlighting the impressive performance of our deep learning model, which was trained solely on simulated datasets generated by Template Learning. We agree that comparing subtomogram averaging of the picked ribosomes with results from models trained on experimental tomograms would complement the assessment.

In response, we have added Supplementary Note 4, which compares subtomogram averaging results based on ribosome annotations from Template Learning, expert-validated annotations, and DeePiCt on the DEF tomograms of EMPIAR-10988. The findings show that subtomogram averaging of ribosomes picked by Template Learning achieved a resolution (~11 Å) identical to that of expert-validated annotations and significantly higher than that of DeePiCt (~15 Å).

Interestingly, further classification of the Template Learning annotations that did not match the expert annotations (initially considered false positives) revealed that approximately 21% of these particles averaged to a clear ribosome-like structure. Refinement of this subset using gold-standard procedures resulted in a resolution of ~ 18 Å, suggesting that Template Learning may have identified valid ribosome particles missed by expert annotation. Further details are provided in Supplementary Note 4.

The approach's generalizability to non-ribosomal targets, particularly within crowded cellular environments and without volta phase plates, requires further demonstration. For example, the authors mention a median F1 score of 0 for FAS annotation on DEF datasets when using only simulated data. Can the authors detail how much fine-tuning with experimental data is necessary when simulated data alone performs poorly? This would provide valuable insights into the practical application of Template Learning for various targets, particularly in crowded cellular environments and without volta phase plates.

We thank the reviewer for this insightful comment. We recognize that accurately detecting non-ribosomal molecules *in situ* without phase plates is a bottleneck in cryo-ET research. Regarding FAS detection in DEF data, while the median F1 score using Template Learning was zero, the method achieved F1 scores of up to 25% in some tomograms (see individual points in the box plots of Fig. 6 in our revised manuscript). This marks an improvement over template matching, which failed to localize any FAS particles [de Teresa-Trueba et al., DeePiCt, Nat Methods 2023]. Additionally, the best median F1 score for FAS picking in DEF tomograms reported for DeePiCt was only 15%.

de Teresa-Trueba et al. highlighted the challenge of annotating FAS due to its scarcity (approximately 75 FAS particles per tomogram on average, compared to 2500 ribosomes) and its shell-like structure, which makes it harder to recognize, especially without phase plates, where data is acquired far from focus and part of the signal becomes delocalized. The authors noted that even with exhaustive manual inspection, less than half the number of FAS particles was annotated in DEF tomograms compared to VPP tomograms (36 FAS particles on average in DEF vs. 75 in VPP), despite similar ribosome annotation counts across both datasets (2500 ribosome annotations on average).

This challenge is evident when visualizing the FAS annotations in both DEF and VPP datasets. The gallery below, displayed using the Eman2 tomography viewer, shows side-by-side comparisons of FAS picks in VPP and DEF tomograms, highlighting the difficulty in annotating FAS, with an even greater challenge in DEF tomograms.

The DEF data contains only 360 annotations across 10 tomograms, with 5 of these tomograms having fewer than 20 annotations, limiting the potential for meaningful fine-tuning experiments. Nonetheless, we found that fine-tuning a model trained via Template Learning with VPP experimental FAS annotations (using all 750 available annotations) and applying it to annotate DEF tomograms (preprocessed with spectrum matching) resulted in a median F1 score of 20%, outperforming previous results (15% reported by DeePiCt).

Regarding the number of particles required for fine-tuning, we can draw insights from our FAS annotation results on VPP data. A model pre-trained on Template Learning simulations and fine-tuned with just 150 (out of 750) experimental FAS annotations improved the median F1 score from 18% to 38%. Notably, de Teresa-Trueba et al. previously reported that training a deep learning model (DeePiCt) with fewer than 600 particles was ineffective. Thus, the ability to fine-tune our model with only 150 particles after pre-training on simulations represents a significant advancement over previous supervised learning methods, which required larger datasets to train from scratch using only experimental annotations. Fine-tuning with 600 out of 750 experimental FAS annotations further increased the median F1 score to 49%, surpassing previous results that used the same number of annotations for training solely on experimental data.

While future versions of Template Learning could incorporate additional considerations for challenging molecules, the current version provides ready-to-use tool—similar to 3D template matching—that can effectively serve many users on its own or as a first step for further fine-tuning with experimental annotations.

VPP tomogram (most annotated)

DEF tomograms (most annotated)

VPP tomogram (least annotated)

DEF tomograms (least annotated)

Caption: Side-by-side galleries of FAS annotations (expert validated) extracted from VPP and DEF tomograms of EMPIAR-10988).

To strengthen the case for broader applicability, the authors could present an additional non-ribosomal example from a cellular environment. Ideally, this example would demonstrate models trained on simulated data either outperforming or performing equally well as those trained on experimental data. This would provide a stronger demonstration of the approach's potential for a broader application.

The answer to this point is combined with the next point.

Template Learning surpasses traditional 3D template matching in picking precision and offers better orientational isotropy for annotating nucleosomes from isolated chromosomes. Applying the trained nucleosome model to publicly available datasets like EMPIAR-10678 or EMPIAR-10179 would strengthen the case for generalizability of the trained model.

We thank the reviewer for these thoughtful suggestions. Indeed, presenting an additional non-ribosomal example from a cellular environment strengthens the demonstration of our approach. While we are confident that Template Learning has potential for broader applications, there is a lack of publicly available datasets with well-annotated benchmarks suitable for evaluating our method.

We carefully reviewed the datasets the reviewer suggested (EMPIAR-10678 and EMPIAR-10179). We found that they either contain VPP tomograms or tomograms collected with a very high defocus (≥ 5 microns), conditions unsuitable for reliable nucleosome annotation based on the team's previous experience [Eltsov et al., NAR, 2018] or subtomogram averaging. The high defocus leads to a loss of resolution, making small structures like nucleosomes appear blobby. Moreover, we were not able to find any nucleosome subtomogram average reported from these datasets, and the VPP tomograms lack the necessary annotations for benchmarking. We also found that this data was collected from the samples with the occurrence of crystallization visible as Bragg reflections that also influence the structure.

However, in response to the reviewer's suggestion, and to demonstrate the applicability of Template Learning in crowded cellular environments, we applied Template Learning to annotate nucleosomes *in situ* using a tomogram obtained by our team with more suitable conditions (defocus around 3 microns and complete vitrification). This tomogram, previously analyzed to study nucleosome dynamics in cells [Eltsov et al. 2018, NAR], was used in two other studies [Harastani et al. 2021, Front. Mol. Biosci.; Harastani et al. 2022, JMB], where 650 nucleosomes were manually annotated with high confidence—a challenging process due to the poor performance of traditional template matching and difficulties in training deep learning models on such a small dataset.

In this new experiment, described in Supplementary Note 5 of our revised manuscript, we compared Template Learning with both supervised deep learning (DeepFinder) and 3D template matching for nucleosome annotation *in situ*. Template Learning produced more accurate and balanced annotations with fewer false positives than the other methods. It reduced the bias toward side-view nucleosomes, achieving a more isotropic angular distribution. Moreover, the subtomogram averages from Template Learning closely resembled the canonical nucleosome structure (PDB 2PYO) more so than those from DeepFinder or 3D template matching.

Further details on this experiment are included in Supplementary Note 5.

References:

Eltsov, M., Grewe, D., Lemercier, N., Frangakis, A., Livolant, F. and Leforestier, A., 2018. Nucleosome conformational variability in solution and in interphase nuclei evidenced by cryo-electron microscopy of vitreous sections. *Nucleic acids research*, 46(17), pp.9189-9200.

Harastani, M., Eltsov, M., Leforestier, A., & Jonic, S. (2021). HEMNMA-3D: cryo electron tomography method based on normal mode analysis to study continuous conformational variability of macromolecular complexes. *Frontiers in molecular biosciences*, 8, 663121.

Harastani, M., Eltsov, M., Leforestier, A. and Jonic, S., 2022. TomoFlow: Analysis of continuous conformational variability of macromolecules in cryogenic subtomograms based on 3D dense optical flow. *Journal of molecular biology*, 434(2), p.167381.

Can Template Learning be effectively used for filamentous macromolecules? If so, testing the model's annotation capability on a dataset like EMPIAR-10989 would be a valuable addition.

Template Learning, as introduced in our manuscript, generates domain-randomized simulations based on a general prior (inductive bias) that models target molecules as flexible and embedded in a highly crowded environment with many non-targeted proteins. We demonstrated that domain randomization-based simulations of flexibility, crowded non-specific background, and acquisition parameters are effective for deep learning on particle picking in cases like ribosomes, FAS, and nucleosomes, where this inductive bias is well-suited.

However, specific scenarios, such as filamentous or membrane proteins, can benefit from more specialized priors for optimal performance. For example, adding additional domain randomization axes to account for how actin organizes into filaments and how these filaments later form bundles would provide valuable context to allow for greater generalization across various filamentous samples.

We recognize that these additional options are beyond the scope of our current manuscript, which is focused on introducing the fundamental concept of domain randomization for cryo-ET particle picking. Nevertheless, this represents a promising direction for future development, and we mentioned this discussion in the Conclusion section of our revised manuscript.

Minor points

For clarity, the text should differentiate between 2D and 3D template matching.

We added to our revised manuscript a clarification that template matching in all our results refers to 3D template matching.

The paper doesn't explicitly address how compositional heterogeneity, more specifically protein-protein interactions, is handled by Template Learning. Clarifying how the method accounts for these variations would be beneficial.

Template Learning allows using multiple structural variations accounting for both compositional and flexible heterogeneity by using different atomic models. As outlined in the Results section under "Accounting for the structural and orientational variabilities," we state: "We use multiple

templates of one target biomolecule to account for compositional or significant conformational variabilities of the target particles."

For example, in our nucleosome pipeline, where we used a nucleosome template bound to the H1 protein (7PEX) alongside five other nucleosome templates without H1 but with varying DNA linker lengths. We have further clarified the reasons behind our choice in our revised manuscript.

In addition to explicitly modeling protein-protein interactions, our experiments suggest that simulating crowded environments with non-specific distractor molecules inherently captures a certain level of protein-protein interaction. This was evident when the Template Learning-trained model successfully detected ribosomes, FAS, and nucleosomes *in situ*, where they likely interacted with other proteins. For more complex protein-protein interactions, we envision that future developments could introduce additional (optional) domain randomization axes to address scenarios where such interactions are critical for optimal performance, such as filamentous or membrane proteins. We have referred to this discussion in the Conclusion of our revised manuscript.

The generation process of the "cytosol masks" used in the study should be described. Understanding how these masks are derived is important for interpreting the results.

Thank you for this valuable suggestion. In response, we have incorporated your feedback along with a suggestion from Reviewer 2. We added a discussion in Supplementary Note 1 that highlights the benefits of using cytosol masking and information on how these masks can be generated both automatically and manually.

We would like to thank Reviewer #4 for reviewing our manuscript and we hope that our revised version meets their expectations.

Reviewer #1 (Remarks to the Author):

Thanks to the authors' responses, these responses partially solve my questions. However, there are still some follow-up concerns:

We thank Reviewer #1 for their comments. Below, we address each comment in detail.

1. While Template Learning falls within the template-matching domain for particle picking, its dependency on high-quality templates could pose limitations, particularly in cases where atomic structures are unavailable—a common situation in *in situ* cryo-ET data analysis. Could the authors provide more insights into how lower-resolution templates (e.g., from low-resolution EM maps) impact model performance and what adjustments, if any, might be necessary for such cases?

We respectfully disagree that dependency on atomic resolution templates poses a limitation for Template Learning. First, in many cases, cryo-electron tomography is employed to map molecules that have already been characterized by *in vitro* structural biology methods such as single-particle cryo-EM, X-ray crystallography or Nuclear Magnetic resonance, providing suitable atomic templates. Second, when only lower-resolution templates are available, such as those from low-resolution EM maps, we proved that Template Learning remained effective. As detailed in the "Using a Volumetric Template" section, we converted a 20 Å resolution volumetric structure into a pseudoatomic model and used it as input. This approach resulted in significantly better performance than traditional template matching, demonstrating that our method accommodates lower-resolution templates without substantial adjustments.

While we expect future users to explore using other lower-resolution templates as input, such studies are beyond the scope of this paper.

2. The method was tested on ribosomes, nucleosomes, and FAS, which are relatively large and structurally defined particles. However, smaller, less-defined particles or highly flexible molecules might perform less robustly with Template Learning. It would be helpful if the authors could discuss the method's limitations for these particle types and clarify if specific adjustments are required for handling diverse macromolecular structures.

While ribosomes are large and structurally defined, picking them *in situ* is challenging due to interactions with other cellular components like proteins and membranes. The fatty acid synthase (FAS) complex, despite its similar radius to ribosomes, presents additional difficulties due to its smaller molecular weight—approximately half that of ribosomes—and shell-like structure, complicating annotation in crowded environments. These challenges were already demonstrated by de Teresa-Trueva et al. [DeePiCt, Nat Methods 2023].

Critically, our study includes nucleosomes, among the smallest non-membrane or non-filamentous structures ever annotated and averaged using cryo-ET, with a molecular weight of approximately 200 kDa—20 times smaller than a ribosome. Successfully annotating nucleosomes underscores Template Learning's capability to handle smaller, more challenging, and

asymmetrical particles. Additionally, nucleosome is dynamic and flexible complex demonstrating aspects of continuous conformational transitions *in vitro* and *in situ* (see for example <https://doi.org/10.1007/s12551-016-0212-z>; <https://doi.org/10.1093/nar/gky670>).

We assert that extremely small, less-defined, or highly flexible molecules pose significant challenges for all particle-picking methods, including template matching and supervised deep learning techniques; these challenges are not unique to our method. To date, cryo-ET data, even with expert human annotations, have not proven effective in reliably studying particles much smaller than a nucleosome. Therefore, adjustments to accommodate these types of particles lie beyond current technological capabilities.

3. Template Learning is proposed as a solution to the orientation biases typically seen in template matching. Could the authors elaborate on how this method specifically mitigates orientation biases, especially when applied to asymmetric particles?

Template Learning eliminates orientation biases by generating training data through simulations where molecules are sampled from uniform angular distributions. This ensures the model is equally exposed to all possible orientations during training, enabling it to recognize particles regardless of their orientation. This was explained in our Results section under “Accounting for the structural and orientational variabilities”.

This approach is particularly advantageous for asymmetric particles, where certain orientations may be underrepresented or harder to detect using traditional methods. Traditional template matching often favors specific orientations, and supervised learning methods can inherit biases from human-annotated data. By ensuring uniform representation of all orientations during training, Template Learning provides a robust and unbiased detection method.

4. I appreciate the authors' acknowledgment of the influence of expert-validated annotations on performance curves, which underlines the need for subtomogram averaging to validate results. However, as shown in Figure 7, the improvement from traditional template matching to Template Learning in subtomogram averaging resolution appears marginal (12.9 Å to 12.8 Å).

The broader context of Figure 7 reveals a significant advancement of Template Learning over template matching. At binning 4, particles picked by traditional template matching yielded a structure at 27 Å resolution, which did not fully resemble the canonical nucleosome. In contrast, particles annotated by Template Learning at the same binning produced a 16.6 Å resolution structure—reaching the Nyquist limit for that binning—and closely resembling the expected nucleosome structure.

Moreover, achieving the 12.9 Å resolution with template matching required rejecting 43% of the particles through 3D classification. Conversely, Template Learning attained this resolution—and slightly better—without discarding any particles, demonstrating a markedly higher initial accuracy in particle picking.

The global Fourier shell correlation (FSC) resolution depends on data homogeneity and consistency. As we mentioned in the answer to the question 2, nucleosomes are flexible and dynamic complexes, and our post-alignment classification of particles picked by Template Learning revealed significant flexibility in the nucleosome DNA entry-exit arms, affecting the global resolution measurement. Local resolution analysis confirmed that the nucleosome core reached significantly higher resolutions, while the flexible DNA arms were resolved at lower resolutions due to their inherent flexibility.

Achieving higher global resolutions would require additional steps like hierarchical classification and focused refinement on flexible regions, which are beyond this study's scope. The key takeaway is that Template Learning provides more accurate initial particle picks, resulting in higher-quality averages without extensive particle rejection or extra processing steps.

We thank you again for reviewing our manuscript and trust that these clarifications address your concerns and enhance the understanding of our work.

Reviewer #2 (Remarks to the Author):

Dear Authors,

all my concerns have been addressed appropriately. I am happy to recommend "Template Learning: Deep Learning with Domain Randomization for Particle Picking in Cryo-Electron Tomography" for publication in Nature Communications.

We thank Reviewer #2 for reviewing our manuscript and we are happy that our revisions addressed their concerns.

Reviewer #3 (Remarks to the Author):

We thank Reviewer #3 for reviewing our manuscript.

Reviewer #4 (Remarks to the Author):

The revisions have addressed my previous concerns and significantly improved the manuscript. I am happy to recommend it for publication.

We thank Reviewer #4 for reviewing our manuscript and we are happy that our revisions addressed their concerns.

Reviewer #1 (Remarks to the Author):

Thanks to the authors' responses, these responses partially solve my questions. However, there are still some follow-up concerns:

We thank Reviewer #1 for their comments. Below, we address each comment in detail.

1. While Template Learning falls within the template-matching domain for particle picking, its dependency on high-quality templates could pose limitations, particularly in cases where atomic structures are unavailable—a common situation in *in situ* cryo-ET data analysis. Could the authors provide more insights into how lower-resolution templates (e.g., from low-resolution EM maps) impact model performance and what adjustments, if any, might be necessary for such cases?

We respectfully disagree that dependency on atomic resolution templates poses a significant limitation for Template Learning. In many cases, cryo-electron tomography is employed to map molecules already characterized by *in vitro* structural biology methods, such as SPA and crystallography, which provide suitable atomic templates. Nonetheless, when only lower-resolution templates are available, such as those from low-resolution EM maps, Template Learning remains effective. We demonstrate this in a proof-of-concept experiment described in the “Using a volumetric template” section of the Results (lines 386–438).

In this experiment, we addressed the scenario suggested by the reviewer. Specifically, we obtained a low-resolution 2-nm template mimicking a sub-tomogram average of ribosomes, referred to as the “volumetric template.” This template was transformed into a pseudoatomic structure following the steps outlined in the Methods section (lines 411–429, 873–889) and subsequently processed to generate synthetic tomographic data.

Our analysis revealed that this approach led to slightly lower F1 scores and precision compared to a model trained with an actual atomic structure and a full set of domain randomization variations (lines 430–438 and Fig. 4A, H). Nevertheless, the F1 score achieved in this experiment represents a 27% improvement relative to traditional template matching, even though both methods utilized a similar prior—a single-volume ribosome template.

As discussed in the Results section (lines 411–429), the main adjustment that might be required relates to the challenges in generating pseudoatomic structure models. Currently, these models may not reproduce the expected contrast when used with physics-based simulators. To address this, we propose an empirical approach (lines 411–426) that allowed contrast tuning in the case of ribosomes (see Extended Data Fig. 4 for illustration and the Methods section for details). This same approach could be applied to other low-resolution templates, and the corresponding scripts will be included in the Template Learning package.

The decay in picking efficiency for the volumetric template can be explained by the removal of high-resolution features (lines 433–435). Additionally, in our proof-of-principle experiment, this volumetric template was used without incorporating shape variations, as would be done in a full domain randomization set (e.g., multiple PDB templates and normal modes). Indeed, benchmarking the volumetric template on ribosome data resulted in a mean F1 score very close to the “single PDB” case (0.76 versus 0.72, respectively). If there is interest from future users, adding deformations to the volumetric template could be explored in future versions of Template Learning. While this represents an exciting avenue for further development, it falls outside the scope of the current manuscript.

2. The method was tested on ribosomes, nucleosomes, and FAS, which are relatively large and structurally defined particles. However, smaller, less-defined particles or highly flexible molecules might perform less robustly with Template Learning. It would be helpful if the authors could discuss the method's limitations for these particle types and clarify if specific adjustments are required for handling diverse macromolecular structures.

While ribosomes are large and structurally defined, picking them *in situ* is challenging due to interactions with other cellular components, such as proteins and membranes. The fatty acid synthase (FAS) complex, despite having a radius similar to ribosomes, presents additional challenges due to its smaller molecular weight—approximately half that of ribosomes—and its shell-like structure, which complicates annotation in crowded environments.

Critically, our study includes nucleosomes, among the smallest non-membrane or non-filamentous structures annotated and averaged using cryo-ET, with a molecular weight of approximately 200 kDa—about 20 times smaller than a ribosome. Successfully annotating nucleosomes underscores Template Learning's capability to handle smaller, more challenging, and asymmetrical particles. Moreover, nucleosomes are dynamic and flexible complexes, often exhibiting continuous conformational transitions both *in vitro* and *in situ* (e.g., <https://doi.org/10.1007/s12551-016-0212-z>; <https://doi.org/10.1093/nar/gky670>).

We assert that extremely small, less-defined, or highly flexible molecules pose significant challenges to all particle-picking methods, including template matching and supervised deep learning approaches. These challenges are not unique to our method. To our knowledge, cryo-ET data, even when assisted by expert human annotations, have not demonstrated robust reliability for studying particles significantly smaller than nucleosomes. Furthermore, even for nucleosomes, annotation requires tomographic data from lamellae or sections approximately 100 nm thick, the routine acquisition of which remains a technical challenge. Recognizing smaller and more flexible molecules inside the crowding cellular environment may require further technical developments in sample thinning methods and in imaging resolution, requirements that are at the current technological frontiers of cryo-ET. On the side of training data simulation, it will necessitate increasing of resolution of synthetic tomograms and exploring structural variations of the target.

Below, we discuss these challenges in detail and propose potential adaptations to our method to enable the annotation of smaller and more flexible molecules:

- Acquiring and processing high-resolution data

Our Template Learning software, enabled by the Parakeet simulator with its MULTEM backend, can generate synthetic data at resolutions of up to 1 Å/pixel. This capability makes our framework directly applicable for training deep learning models on high-resolution cryo-ET data, as they become available. However, current limitations, including alignment errors, narrow fields of view, dose damage, and low contrast, restrict the routine acquisition of such high-resolution data. Future developments—such as laser phase plates, improved hardware stability (e.g., cryo-TEM grid holder and microscope precision), enhanced sample preparation techniques, and better software for alignment—will likely address these challenges. Our Template Learning software is prepared to accommodate these advancements.

- Conformational flexibility representation in simulated training data

For ribosomes, FAS, and nucleosomes, six distinct PDB templates with conformational variations simulated via normal mode analysis (NMA) were sufficient for the presented proof-of-principle of Template Learning. Given the imperfections in current cryo-ET data quality and the lack of exhaustive annotations, it remains unclear whether the template set (6 PDBs + NMA flexibility) fully captured all structural variations. For highly flexible molecules, where data quality permits detection, simulations might need to sample a broader range of conformational changes. Template Learning allows users to adjust the number of initial models, the amplitudes of NMA, and the number of generated structures. However, NMA simulates harmonic oscillations around an energy minimum and may not capture extensive conformational changes (<https://doi.org/10.1016/j.str.2005.02.002>). Atomic models generated by molecular dynamics simulations or AlphaFold predictions could complement NMA to address these limitations albeit at the cost of a very significant additional computational overhead.

Based on the recommendation of the reviewer, we have incorporated the following discussion on the limitations in the Conclusion section of our manuscript:

We demonstrate that Template Learning is effective for annotating relatively large and structurally well-defined particles, such as ribosomes, FAS, and nucleosomes. However, smaller, less-defined, or highly flexible molecules may still present unique challenges. These challenges stem from inherent limitations in cryo-ET data quality and the increased complexity required to adequately represent such particles in simulations. Future advancements in cryo-ET hardware and software are likely to enable higher-resolution data acquisition. For flexible molecules, incorporating more diverse structural templates, potentially derived from molecular dynamics simulations or AlphaFold predictions, could significantly enhance Template Learning's applicability, albeit at the cost of additional computational overhead."

3. Template Learning is proposed as a solution to the orientation biases typically seen in template matching. Could the authors elaborate on how this method specifically mitigates orientation biases, especially when applied to asymmetric particles?

Template Learning eliminates orientation biases by generating training data through simulations where molecules are sampled from uniform angular distributions. This ensures the model is equally exposed to all possible orientations during training, enabling it to recognize particles regardless of their orientation. We added this explanatory sentence into the “Accounting for the structural and orientational variabilities” section (lines 132–140). This approach is particularly advantageous for asymmetric particles, where certain orientations may be underrepresented or harder to detect using traditional methods. Traditional template matching often favors specific orientations, and supervised learning methods can inherit biases from human-annotated data. In our manuscript, the orientation bias inherent to template matching is demonstrated using the example of the nucleosome, a non-symmetric, disc-like object. Applying template matching followed by sub-tomogram averaging revealed an uneven distribution of nucleosome orientations, with a predominant side-view representation (lines 606–631 and Extended Data Fig. 8). Our results on nucleosome picking confirm that the model trained on synthetic data achieves rotational invariance and can efficiently pick orientations of nucleosomes that are not detectable by template matching (Results section “Template Learning offers higher Precision and better orientational isotropy than template matching on nucleosome picking,” lines 537–546, and Supplementary Note 5).

By ensuring uniform representation of all orientations during training, Template Learning provides a robust and unbiased detection method.

4. I appreciate the authors' acknowledgment of the influence of expert-validated annotations on performance curves, which underlines the need for subtomogram averaging to validate results. However, as shown in Figure 7, the improvement from traditional template matching to Template Learning in subtomogram averaging resolution appears marginal (12.9 Å to 12.8 Å).

The broader context of Figure 7 reveals a significant advancement of Template Learning over template matching. At binning 4, particles picked by traditional template matching yielded a structure at 27 Å resolution, which did not fully resemble the canonical nucleosome. In contrast, particles annotated by Template Learning at the same binning produced a 16.6 Å resolution structure—reaching the Nyquist limit for that binning—and closely resembling the expected nucleosome structure.

Moreover, achieving the 12.9 Å resolution with template matching required rejecting 43% of the particles through 3D classification. Conversely, Template Learning attained this resolution—and slightly better—without discarding any particles, demonstrating a markedly higher initial accuracy in particle picking.

The global Fourier shell correlation (FSC) resolution depends on data homogeneity and consistency. As we mentioned in the answer to the question 2, nucleosomes are flexible and dynamic complexes, and our post-alignment classification of particles picked by Template Learning revealed significant flexibility in the nucleosome DNA entry-exit arms, affecting the global resolution measurement. Local resolution analysis confirmed that the nucleosome core reached significantly higher resolutions, while the flexible DNA arms were resolved at lower resolutions due to their inherent flexibility.

Achieving higher global resolutions would require additional steps like hierarchical classification and focused refinement on flexible regions, which are beyond this study's scope. The key takeaway is that Template Learning provides more accurate initial particle picks, resulting in higher-quality averages without extensive particle rejection or extra processing steps.

We thank you again for reviewing our manuscript and trust that these clarifications address your concerns and enhance the understanding of our work.

Reviewer #2 (Remarks to the Author):

Dear Authors,

all my concerns have been addressed appropriately. I am happy to recommend "Template Learning: Deep Learning with Domain Randomization for Particle Picking in Cryo-Electron Tomography" for publication in Nature Communications.

We thank Reviewer #2 for reviewing our manuscript and we are happy that our revisions addressed their concerns.

Reviewer #3 (Remarks to the Author):

We thank Reviewer #3 for reviewing our manuscript.

Reviewer #4 (Remarks to the Author):

The revisions have addressed my previous concerns and significantly improved the manuscript. I am happy to recommend it for publication.

We thank Reviewer #4 for reviewing our manuscript and we are happy that our revisions addressed their concerns.